# Optimization of Carbon Dioxide Utilization: Simulation-Based Analysis of Reverse Water Gas Shift Membrane Reactors

**DOI:** 10.3390/membranes15040107

**Published:** 2025-04-01

**Authors:** Putri Permatasari, Manabu Miyamoto, Yasunori Oumi, Yogi Wibisono Budhi, Haroki Madani, Teguh Kurniawan, Shigeyuki Uemiya

**Affiliations:** 1Department of Materials Science and Processing, Gifu University, Gifu 501-1113, Japan; putri.permatasari.s4@s.gifu-u.ac.jp; 2Department of Chemistry and Biomolecular Science, Gifu University, Gifu 501-1113, Japan; miyamoto.manabu.v5@f.gifu-u.ac.jp (M.M.); oumi.yasunori.x2@f.gifu-u.ac.jp (Y.O.); 3Department of Chemical Engineering, Bandung Institute of Technology, Bandung 40116, Indonesia; y.wibisono@itb.ac.id; 4Research Center for Chemistry, National Research and Innovation Agency (BRIN), South Tangerang 15314, Indonesia; haroki.madani@brin.go.id; 5Department of Chemical Engineering, Sultan Ageng Tirtayasa University, Serang 42435, Indonesia; teguh@untirta.ac.id

**Keywords:** membrane reactor, RWGS, CO_2_ conversion, computational simulation, process optimization

## Abstract

This study focuses on optimizing the Reverse Water Gas Shift (RWGS) reaction system using a membrane reactor to improve CO_2_ conversion efficiency. A one-dimensional simulation model was developed using FlexPDE Professional Version 8.01/W64 software to analyze the performance of ZSM-5 membranes integrated with 0.5 wt% Ru-Cu/ZnO/Al_2_O_3_ catalysts. The results show that the membrane reactor significantly outperforms the conventional Packed Bed Reactor by achieving higher CO_2_ conversion (0.61 vs. 0.99 with optimized parameters), especially at lower temperatures, due to its ability to remove H_2_O and shift the reaction equilibrium selectively. Key operational parameters, including temperature, pressure, and sweep gas flow rate, were optimized to maximize membrane reactor performance. The ZSM-5 membrane showed strong H_2_O selectivity, with an optimum operating temperature of around 400–600 °C. The problem is that many reactants permeate at higher temperatures. Subsequently, a Half-MPBR design was introduced. This design was able to overcome the reactant permeation problem and increase the conversion. The conversion ratios for PBR, MPBR, and Half-MPBR are 0.71, 0.75, and 0.86, respectively. This work highlights the potential of membrane reactors to overcome the thermodynamic limitations of RWGS reactions and provides valuable insights to advance Carbon Capture and Utilization technologies.

## 1. Introduction

Significant reductions in carbon dioxide (CO_2_) and other greenhouse gas emissions are essential to address global climate change [1,2,3,4,5]. The technology for capturing, utilizing, and storing carbon dioxide (CCUS) plays a significant role in mitigating global CO_2_ emissions. It facilitates the transformation of CO_2_ into fuels and chemical products, in addition to enabling long-term geological storage, thus contributing to the fight against climate change [6,7,8,9,10]. Creative applications of CO_2_ hold the promise of establishing a multi-billion-dollar market for CO_2_ conservation; however, they need substantial funding, policy backing, and research assistance [8,9,11,12,13].

Carbon Capture and Utilization (CCU) through the conversion of CO_2_ into syngas through the Reverse Water Gas Shift (RWGS) reaction is a potential solution in mitigating climate change, although the costs are still higher than Carbon Capture and Storage (CCS) [14,15,16,17,18,19,20]. The RWGS reaction (CO_2_ + H_2_ ⇌ CO + H_2_O) is an endothermic process that converts CO_2_ and H_2_ into CO and H_2_O. This reaction is particularly important as it produces syngas, which serves as a versatile feedstock for various chemical processes, including the production of synthetic fuels and valuable chemicals [21,22,23,24,25,26,27,28]. Green hydrogen production methods such as water splitting using solar energy or electrolysis, which are also being widely developed, make RWGS more feasible as a sustainable CO_2_ consumption strategy [29,30,31,32]. Kinetic limitations at low temperatures, poor reaction efficiency, and insufficient selectivity are the primary issues with RWGS [17,33,34,35,36].

RWGS reactions rely heavily on catalysts [22,37,38]. Recent research has focused on developing novel catalysts to enhance RWGS performance. For example, studies have explored the use of bimetallic catalysts, such as Ni-Fe and Pt-Co, which have shown improved activity and selectivity compared to traditional catalysts [39,40,41]. The low cost of Cu/ZnO/Al_2_O_3_ catalysts makes them widely employed, but deactivation at high temperatures is a problem [17,33,42,43]. According to recent developments, adding 0.5 weight percent Ru can greatly increase CO_2_ conversion and catalyst stability without resulting in the byproduct of methane [33,44]. Achieving a compromise between rapid reaction rate and good selectivity also requires optimizing operational parameters including temperature, pressure, and the H_2_/CO_2_ ratio [17,18,33,34,45,46,47].

Membrane reactors maximize CO_2_ conversion and product selectivity by integrating reaction and separation into a single unit [23,45,46,48,49,50,51]. By removing H_2_O from the reaction system, this method can shift the reaction balance more positively. The integration of membranes in catalytic reactors also helps overcome thermodynamic limitations and improve product selectivity [45,46,49,50,52,53,54]. Optimizing the system is essential to lowering costs and increasing efficiency, particularly when using membrane reactors. In RWGS, membrane reactors can save operating costs and improve energy efficiency by up to 30% compared to conventional reactors [55,56]. Various types of membranes have been investigated for RWGS applications, including zeolite membranes (such as ZSM-5), perovskite membranes, and polymer membranes. Each type offers unique advantages in terms of selectivity, permeability, and stability under RWGS conditions [57,58]. ZSM-5 membranes have the advantages of a unique porous structure with narrow cross-shaped channels, a pore size that allows selectivity to certain sized molecules, plus the ability to separate mixtures based on hydrophilicity that can be adjusted from hydrophobic to highly hydrophilic [59,60].

In system optimization, computational simulation is essential because it makes it possible to assess different operating conditions without the need for costly physical trials [48,51,61,62,63]. This study will employ a simulation to optimize the RWGS system. By integrating experimental data with advanced modeling techniques, we can develop a comprehensive understanding of the reaction dynamics and transport phenomena within the membrane reactor [64]. This covers sensitivity analysis and reaction kinetics computations. By combining catalyst and membrane data from earlier research and employing computational methods to identify ideal settings, this work seeks to improve the RWGS system using a membrane reactor. By developing a data-based simulation model of the ZSM-5 membrane and 0.5 wt% Ru-Cu/ZnO/Al_2_O_3_ catalyst, the primary goal is to increase the CO_2_ conversion efficiency [23,33]. This approach will hopefully support mitigating climate change and the shift to a low-carbon economy by assisting in the development of novel technologies to lower CO_2_ emissions and boost energy efficiency.

## 2. Modeling and Simulation

The main equation used in this study is shown in the equation below. A series of equations are used to calculate the reaction rate (Equations (1)–(5)), including the reaction rate equation, the equation for calculating the reaction rate constant, the equation for determining the adsorption constant of each component, the denominator function, which represents the effect of adsorption, and the equation for calculating the reaction equilibrium constant [33].(1)$$R_{RWGS}=k_{r}\left(P_{{CO}_{2}}-\frac{P_{H_{2}O}P_{CO}}{K_{RWGS}P_{H_{2}}}\right)\frac{1}{{DEN}^{2}}$$(2)$$k_{r}=k_{ref}\exp\left(\frac{E_{a}}{R}\left[\frac{1}{T_{ref}}-\frac{1}{T}\right]\right)$$(3)$$DEN=1+K_{{CO}_{2}}P_{{CO}_{2}}+K_{CO}P_{CO}+\frac{K_{H_{2}O}P_{H_{2}O}}{P_{H_{2}}}$$(4)$$K_{i}=K_{i,ref}\exp\left(\frac{{\Delta H}_{i}}{R}\left[\frac{1}{T_{ref}}-\frac{1}{T}\right]\right)$$(5)$$K_{RWGS}=exp\left(\frac{-\Delta G}{RT}\right)$$

Equations (6) and (7) are then used to compute permeation on the ZSM 5 membrane, as well as the permeance frequency factor for each component.(6)$$Q_{i}=J_{i}\times {MA}_{2}\times \left(P_{i,reac}-P_{i,perm}\right)$$(7)$$J_{i}=J_{0,i}e^{\frac{-E_{a,j}}{RT}}$$

Equations (8)–(16) show the material balance employed in the calculation. Equations (8)–(11) are for the reaction side, whereas Equations (12)–(16) are for the permeation side. In the equation, *Q_i_* represents the component i’s permeation, whereas in the PBR model with no permeation, *Q_i_* equals 0.(8)$${CO}_{2,reac}:\varepsilon \frac{𝜕F_{{CO}_{2}}}{𝜕x}=-R_{RWGS}\rho_{cat}A_{in}(1-\varepsilon )\;-\;Q_{{CO}_{2}}$$(9)$$H_{2,reac}:\varepsilon \frac{𝜕F_{H_{2}}}{𝜕x}=-R_{RWGS}\rho_{cat}A_{in}\left(1-\varepsilon \right)\;-\;Q_{H2}$$(10)$${CO}_{reac}:\varepsilon \frac{𝜕F_{CO}}{𝜕x}=R_{RWGS}\rho_{cat}A_{in}\left(1-\varepsilon \right)\;-\;Q_{CO}$$(11)$${H_{2}O}_{reac}:\varepsilon \frac{𝜕F_{H_{2}O}}{𝜕x}=R_{RWGS}\rho_{cat}A_{in}\left(1-\varepsilon \right)-Q_{H_{2}O}$$(12)$${CO}_{2,perm}:\varepsilon \frac{𝜕F_{{CO}_{2}}}{𝜕x}=Q_{{CO}_{2}}$$(13)$${H_{2}}_{,perm}:\varepsilon \frac{𝜕F_{H_{2}}}{𝜕x}=Q_{H2}$$(14)$${CO}_{perm}:\varepsilon \frac{𝜕F_{CO}}{𝜕x}=Q_{CO}$$(15)$${H_{2}O}_{perm}:\varepsilon \frac{𝜕F_{H_{2}O}}{𝜕x}=Q_{H_{2}O}$$(16)$$Ar:\varepsilon \frac{𝜕F_{Ar}}{𝜕x}=0$$

This system’s heat balance computation considers five major aspects. First, the heat balance on the permeation side considers the energy transfer between the permeate gas and the membrane (Equation (17)). Second, the balance on the reaction side considers the heat created or absorbed during the reaction, as well as the transfer to the membrane and reactor wall (Equation (18)). In addition, the calculation includes the overall heat transfer coefficient, which represents the system’s thermal efficiency (Equation (19)). The heat transfer coefficient across the membrane (*U*_1_) measures the efficiency of energy transmission from the reaction zone to the permeation side, whereas the heat transfer coefficient through the reactor wall (*U*_2_) measures heat loss to the environment (Equations (20) and (21)) [65,66].(17)$$\left(F_{H2O}C_{pm,H2O}+F_{Ar}C_{pm,Ar}\right)\frac{dT_{perm}}{dx}=U_{1}{MA}_{2}\left(T_{reac}-T_{perm}\right)+(\;Q_{CO2}C_{pm,CO2}+\;{Q_{H2}C_{pm,H2}+Q_{CO}C_{pm,CO+}Q_{H2O}C_{pm}}_{H2O}\left(T_{reac}-T_{perm}\right))$$(18)$$\left(F_{CO2}C_{pm,CO2}+F_{H2}C_{pm,H2}+F_{CO}C_{pm,CO}+F_{H2O}C_{pm,H2O}\right)\frac{dT_{reac}}{dx}=\;A_{in}\rho_{cat}\left(-r_{RWGS}\right)\left({\Delta H}_{R,T}\right)+U_{1}{MA}_{2}\left(T_{perm}-T_{reac}\right)+U_{2}{MA}_{3}\left(T_{wall}-T_{shell}\right)$$(19)$$\frac{h_{w}d_{p}}{\lambda_{F}}=\left(\frac{d_{p}}{d}\right)\left(\frac{\lambda_{er}}{\lambda_{F}}\right)\left\{{a_{1}}^{2}+\frac{\Phi \left(b\right)}{\xi }\right\}$$(20)$$\frac{1}{U_{1}Z{MA}_{1}}=\frac{1}{h_{in}Z{MA}_{1}}+\frac{{(d}_{2}-d_{1})/2}{\lambda_{al}A_{m1}}+\frac{1}{h_{out}Z{MA}_{2}}$$(21)$$\frac{1}{U_{2}Z{MA}_{3}}=\frac{1}{h_{out}Z{MA}_{3}}+\frac{{(d}_{4}-d_{3})/2}{\lambda_{sus}A_{m2}}$$

Next, a thermodynamic equation is used (Equations (22)–(24)).(22)$$\Delta H_{reaction}^{{}^{\circ}}=\sum {\Delta H}_{f(product)}^{{}^{\circ}}-\sum {\Delta H}_{f(reactan)}^{{}^{\circ}}$$(23)$$\Delta S_{reaction}^{{}^{\circ}}=\sum {\Delta S}_{\left(product\right)}^{{}^{\circ}}-\sum {\Delta S}_{\left(reactan\right)}^{{}^{\circ}}$$(24)ΔG=ΔH−TΔS

Next, calculations are made to determine the impact of withdrawing water from the system. These calculations rely on three main equations: the fluid continuity equation (Equation (25)) to maintain mass balance in the system, the reaction equilibrium equation (Equation (26)) to account for changes caused by water removal, and an equation to calculate the amount of water removed from the system during the process (Equation (27)). The combination of these equations is used to calculate the equilibrium conversion when water is eliminated, allowing for examination of the effect on reaction efficiency.(25)$$\rho_{1}V_{1}=\rho_{2}V_{2}$$(26)$$K_{RWGS}=\frac{\left(X\right)\left(MX\right)}{\left(1-X\right)\left(ratio-X\right)}$$(27)$${H_{2}O}_{removed}=\frac{{H_{2}O}_{Perm}}{{{{H_{2}O}_{reac}+H}_{2}O}_{perm}}$$

To test whether a one-dimensional model can reflect the system, the Péclet number and related variables are carefully determined. This computation is performed using Equations (28)–(35), which involve several parameters that influence the validity of the one-dimensional approach.(28)$$Pe=\frac{Zu}{D}$$(29)$$D_{er}=\frac{u_{s}d_{p}}{11\left[1+19.4{\left(\frac{d_{p}}{D_{i}}\right)}^{2}\right]}$$(30)$$Da=\frac{k_{f}L_{0}}{u_{z}}$$(31)$${Pe}_{i}=\frac{D_{i,r}L_{0}}{u_{i}R_{0}^{2}}$$(32)$$\theta_{i}=\frac{u_{z}}{P_{i}L_{0}}$$(33)$$\xi =\frac{c_{tot}}{K_{E}}=\frac{p_{tot}}{K_{E}RT}$$(34)$${Pe}_{crit,i=\frac{{a\varepsilon }^{b}}{\left(1+{c\xi }^{d}{Da}^{f}+{g\xi }^{h}{Da}^{j}\right)\left(1+{k\xi }^{l}\theta^{m}\right)}}$$(35)$${Pe}_{crit}=max\left({Pe}_{crit,a}{Pe}_{crit,b}{Pe}_{crit,c}{Pe}_{crit,d}\right)$$

Then, other pertinent equations, such as component ratio computations and membrane selectivity equations, are applied.(36)$${Ratio}_{i}=\frac{F_{i,perm\;side}}{F_{i,reac\;side}+F_{i,perm\;side}}$$(37)$${Ratio}_{i}=\frac{F_{i,reac\;side}}{F_{i,reac\;side}+F_{i,perm\;side}}$$(38)αH2Oi=xH2O,permeatexH2O,retentatexi,permeatexi,retentae

In addition, some additional equations were used in the development of the simulation system to improve the accuracy and validity of the model. Further details regarding the formulation and implementation of the equations can be obtained by contacting the author.

### 2.1. System Development for RWGS Reaction in PBR

The development of a system for the RWGS reaction was initiated using a 0.5 wt% Ru-Cu/ZnO/Al_2_O_3_ catalyst data (Table 1), based on the research conducted by Zhuang et al. in 2019 [33]. By leveraging this information and incorporating it into system equations, a comprehensive simulation model was constructed. This model served as a foundation for understanding the complex interplay between catalyst characteristics and reaction kinetics, enabling the optimization of the RWGS process in subsequent stages of the research. Equations (1)–(4) are taken directly from the journal. The reaction conditions for each condition in the experiment vs. simulation are specifically stated in Table 2. Following that, additional data are required to ensure the simulation process runs effectively (Table 3).

In the reference journal, the reactor’s inner diameter was not specified, despite being a crucial parameter for simulation. Therefore, it had to be determined independently. Based on the available data, the reactor uses a 1/4-inch stainless steel tube with a K-type thermocouple (1/8 inch or 3.175 mm diameter) directly installed inside. The tube’s inner diameter depends on the wall thickness (0.035”, 0.049”, or 0.065”), calculated using the following formula: outer diameter minus twice the wall thickness. The results show that the inner diameter ranges from 3.05 to 4.57 mm. To ensure proper thermocouple installation, a 4.5 mm diameter was chosen as the optimal value, providing sufficient space without compromising mechanical stability.

The simulation was developed using validated experimental data to ensure accuracy in representing real conditions. Once the model was validated, parameters such as temperature, pressure, and flow rates were adjusted to explore optimal conditions. This optimization provides a foundation for further simulations to enhance catalytic system performance.

### 2.2. System Development for RWGS Reaction in MPBR (Ideal H_2_O-Selective Membrane)

The reaction conditions utilized in this section are listed in Table 4. The values that are not displayed are the same as those in the preceding section. Figure 1 illustrates the system configurations employed in this study. It presents two distinct reactor types: the Packed Bed Reactor (PBR) and the Membrane Packed Bed Reactor (MPBR). Table 5 provides a comprehensive overview of the critical dimensions of each reactor design, including length, surface area, and volume. To ensure a fair and accurate comparison between the different reactor configurations, all systems have been carefully designed to maintain a uniform catalyst density throughout. This standardization allows for a more precise evaluation of the reactors’ performance and efficiency in the RWGS reaction.

In this section, we will examine and compare the designs of different reactor systems. For MPBR, we have developed two configurations: one with the reaction occurring on the shell side (MPBR-Out) and another with the reaction on the tube side (MPBR-In). These configurations allow us to evaluate the impact of reaction placement on overall system performance. In the MPBR-In setup, the reaction takes place within the tube, while in the MPBR-Out configuration, the reaction occurs in the shell surrounding the tube. It is important to note that these design variations not only affect the placement of the material balance but also require adjustments to the energy balance equations for each specific condition. This ensures accurate modeling of heat transfer and reaction kinetics in both configurations. For a visual representation of these reactor designs, please refer to Figure 2.

In the next step, we examine the impact of different flow arrangements by implementing both co-current and counter-current configurations. The inlet sweep boundary condition is shifted to Z = L, and a negative sign is introduced on the right-hand side of the sweep side differential equation (Fe, Ff, and Tgt). These modifications are aimed at exploring how the direction of the sweep flow influences the overall reaction performance and equilibrium. An illustration of these configurations is provided in Figure 2.

Next, we focus on parameter changes for the RWGS reaction in an MPBR using an ideal H_2_O-selective membrane to determine the optimal operating conditions. First, the relationship between the equilibrium conversion and the amount of water removed from the system is calculated using Equation (26). The *M* value represents the fraction of water remaining in the system, where *M* = 1 indicates no water removal (0% removed) and *M* = 0 indicates complete water removal (100% removed). As more water is removed from the system, the *M* value decreases proportionally, with a reduction of 0.1 in *M* value for every 10% increase in water removal. This systematic relationship provides a clear quantitative measure of water removal efficiency in the RWGS reaction system. The calculated values are then compared with the results of the simulation.

### 2.3. System Development for RWGS Reaction in MPBR (ZSM-5 Membrane)

This system integrates research on catalysts and membranes from two key studies: Zhuang et al. (2019) [33], which focuses on the 0.5 wt% Ru-promoted Cu/ZnO/Al_2_O_3_ catalyst, and Sakai et al. (2022) [23], which examines zeolite membrane reactors. Table 6 presents the experimental conditions and dimensions used in the simulation. In this system, the ZSM-5 membrane allows permeation of all components (CO_2_, H_2_, CO, and H_2_O), requiring careful consideration of each component’s permeation capabilities. The membrane permeation characteristics for each component were determined through fitting parameters that align experimental and simulation results. Pre-exponential factors and activation energies were established for each component (Table 6), with permeation rates calculated using Equation (7) following the Arrhenius approach. This method enables permeation calculations across various temperatures, with the results validated through comparison between experimental and simulation data.

It is essential to highlight the significant differences between the two MPBR ideal systems studied in this research. The two ideal MPBR systems discussed exhibit crucial distinctions. The first system, outlined in Section 2.2, employs an idealized H_2_O-selective membrane, ignoring the permeation of other components. This approach leads to a unique determination of activation energy and frequency factor values. In contrast, the second system, detailed in Section 3.3, is specifically designed for the ZSM-5 membrane. Its activation energy and frequency factor values are meticulously calculated, considering the permeation of additional components such as CO_2_, H_2_, and CO. Although both systems (the system in Section 2.2 and one of the systems in Section 2.3) are labeled as “ideal”, the disparities in their activation energy and frequency factor values result in markedly different performance characteristics, setting them apart in terms of behavior and efficiency. The systems also differ in several key parameters: reactor length (8 cm vs. 9 cm), catalyst mass (0.5 g vs. 3.2 g), total feed flow rates (50 mL/min vs. 12 mL/min), reactant composition (1/4 vs. 1/3 CO_2_/H_2_), membrane permeability characteristics (single vs. multiple component permeation), and sweep gas flow rates (50 mL/min vs. 5 mL/min). These substantial variations in system parameters make direct performance comparisons between the two systems neither meaningful nor valid.

Moving forward, the simulation aims to represent the actual system using real catalyst and membrane data, necessitating thorough validation of the model’s accuracy. The implementation of one-dimensional (1D) models for reactor systems requires meeting specific validation criteria. To verify the model’s feasibility, several verification steps were performed, including the application of the Péclet number criterion developed by Lundin et al. [67]. While this verification method was initially designed for hydrogen-permeating membranes, it has been adapted with appropriate modifications to validate our system. The mathematical framework for this verification process is outlined in Equations (28)–(35), which provide the necessary calculations to confirm the model’s validity.

As in Section 2.2, manual calculations were performed to evaluate how water removal from the reaction side effects the CO_2_ conversion across different temperatures, but with parameters adjusted for this section’s conditions. These calculations serve as a validation benchmark, establishing maximum theoretical conversion limits for different water removal ratios. By comparing simulation results against these theoretical limits, we can verify that the model maintains physical accuracy and does not predict conversions beyond thermodynamic equilibrium constraints.

Subsequently, several efforts were made to study and improve the system’s efficiency. These efforts included optimizing reaction conditions, modifying membrane properties, and exploring different reactor configurations to enhance the overall performance of the RWGS reaction system. The reactor design comparison examines three distinct configurations as shown in Figure 3. The Half-MPBR represents an innovative design where the membrane begins partway through the reactor rather than at the entrance. All three reactor configurations were evaluated under identical reaction conditions, which are detailed in Table 7, ensuring a fair comparison of their performance characteristics.

## 3. Results and Discussion

### 3.1. RWGS Reaction System in PBR

The simulation model developed for the RWGS reaction using catalyst data demonstrates a strong correlation with experimental results. Comparisons between simulated and experimental data were made across three key parameters: conversion versus residence time (W/F), H_2_/CO_2_ ratio, and temperature (Figure 4). The model shows good agreement with the experimental data obtained by Zhang et al., accurately representing the reaction behavior under various conditions. While some minor deviations are observed, the overall performance of the simulation indicates its reliability in predicting RWGS reaction outcomes. This robust model provides a valuable tool for further optimization and analysis of the RWGS process.

Next, the research focuses on optimizing the RWGS system. Figure 5a illustrates the relationship between temperature and conversion for two configurations: the initial and the optimized configuration. The initial configuration represents the setup used in previous experiments, while the optimized configuration is based on RWGS reaction principles that show better performance at lower pressures and reduced volumetric flow rates. Changes can be seen in Table 8, while Figure 5b shows the conversion profile against W/F at the selected temperature of 400 °C. Although system efficiency improves with these parameter adjustments, thermodynamic limitations remain a challenge. This emphasizes the need for membrane integration to overcome these limitations. By incorporating a membrane, a Membrane Packed Bed Reactor system can be developed using this optimized configuration. This updated model will be used in the development stage of membrane reactors with ideal water selectivity.

### 3.2. System for RWGS Reaction in MPBR (Ideal H_2_O-Selective Membrane)

#### 3.2.1. In vs. Out Arrangement

Figure 6 presents a comparison of reactor conversion versus W/F, alongside water pressure and temperature profiles throughout the reactors. Due to the small size of the reactor and the same catalyst density, the variations in conversion are minimal, indicating that reactor placement has negligible impact. In more detail, in a conventional Packed Bed Reactor (PBR), conversion increases with W/F but remains limited by equilibrium. By contrast, the Membrane Packed Bed Reactor (MPBR) achieves higher conversion through water removal, although showing only slight differences between the MPBR-In and MPBR-Out configurations. This minimal disparity stems from the small reactor scale, as well as the similar densities employed. Additionally, while the water pressure is slightly higher in the MPBR-Out configuration, it has an insignificant effect under these conditions.

Finally, because Reverse Water Gas Shift (RWGS) is endothermic, the slight oscillation observed in the temperature profile (Figure 6c) is likely due to localized thermal effects at the reactor inlet, where the reaction rate is highest. At this point, the rapid CO_2_ hydrogenation could cause minor temperature fluctuations before reaching a steady-state profile. However, for reactions that are on the shell side (Out configuration), the effect is compensated by continuous heating from the reactor walls, leading to faster temperature stabilization further along the reactor. Vice versa, because in the In configuration (the reaction is on the tube side) the distance from the heat source (reactor wall) is further, the temperature drop is greater, although not that big.

Additionally, most of the reactor exhibits a stable temperature profile, with oscillations occurring only in a small region near the inlet. In experimental setups, temperature sensors are typically placed in the middle of the packed bed, which may explain why this fluctuation is not commonly observed in measured data. The observed oscillations remain within a narrow range, indicating that they do not significantly affect system performance.

However, there is a possibility of numerical issues. Using the current program (FlexPDE Professional Version 8.01/W64), we obtained the results presented in Figure 6. We have not tested other software. In the future, we may compare different systems to determine whether the observed results represent true oscillations or are artifacts of numerical errors.

#### 3.2.2. Flow Regulation (Co-Current vs. Counter-Current)

In the co-current arrangement, the reactor achieves higher CO_2_ conversion at lower weight-to-flow (W/F) ratios and more quickly reaches equilibrium. By contrast, the counter-current arrangement excels at higher W/F ratios, resulting in superior CO_2_ conversion under those conditions. In terms of water transport, co-current flow exhibits a gradual increase in H_2_O pressure across both the reaction and permeation sides, while counter-current flow produces a sharp H_2_O pressure peak near the reactor inlet, creating a notable pressure difference early on. The operating temperature is tightly controlled between 723.13 and 723.16 K, causing the co-current flow to experience larger temperature fluctuations at the reactor entrance. Meanwhile, the counter-current system remains more stable and reaches thermal equilibrium after approximately 1 cm of reactor length. These findings are displayed in Figure 7.

#### 3.2.3. Analysis of Parameter Changes in MPBR with Ideal H_2_O-Selective Membrane

Figure 8 presents the relationship between equilibrium conversion and water removal from the RWGS reaction system. The calculations demonstrate that conversion efficiency increases proportionally with the amount of water removed from the system. This relationship serves as a fundamental reference point for validating simulation results, providing a theoretical framework to assess the system’s performance under various water removal conditions. Note here that Figure 8 illustrates the theoretical upper limit of conversion, assuming complete removal of water from the system. However, water removal occurs through membrane permeation, which is driven by partial pressure differences. As a result, complete water removal cannot be practically achieved, and the conversion does not reach exactly 1. Therefore, although Figure 8 serves as a reference to illustrate the maximum possible equilibrium shift, the actual performance of the system is limited by the ability of the membrane to selectively remove water under realistic operating conditions.

The next phase of analysis focuses on evaluating system conversion across various temperatures. A comprehensive comparison of CO_2_ conversion was conducted across a temperature range of 0–1000 °C to assess the performance differences between PBR and MPBR systems. The results, shown in Figure 9a, demonstrate that MPBR significantly outperforms PBR by surpassing thermodynamic equilibrium limitations. The amount of water removed during the process, calculated using Equation (27), is illustrated in Figure 9b. In the conventional PBR system, equilibrium conversion is reached at 400 °C and plateaus, thereafter, indicating a clear thermodynamic constraint. However, the MPBR system achieves equilibrium conversion at a lower temperature of 350 °C, highlighting its superior efficiency. Regarding water separation effectiveness, the process shows significant improvement up to 400 °C, where it stabilizes at 55% removal due to maximum membrane permeation capacity. Beyond this temperature, the partial pressure difference approaches zero, as shown in Figure 9c, which limits further separation and conversion capabilities. This limitation suggests that additional optimization strategies are needed to achieve higher conversion rates in the system.

To address this limitation, we investigated the effect of modifying water permeance on system performance. We evaluated how changing the permeance parameter (*J*) from its baseline value of 3.27 × 10^−7^ mol/m^2^·s Pa affected CO_2_ conversion. Despite increasing water permeance, we observed no meaningful improvement in system performance (see Figure 10a). The lack of improvement was due to unchanged partial pressure differences across the membrane, which meant water removal and conversion rates remained constant regardless of increased permeance values (see Figure 10b).

Another optimization approach focuses on modifying the permeate side sweep gas flow rate. This study evaluated CO_2_ conversion by varying the sweep gas flow rate from 1 to 100 times the reference conditions (50 mL/min for both feed gas and sweep gas) (Figure 11). The results showed a significant increase in conversion when the sweep gas flow rate was increased by 5–15 times the reference value, while further increases yielded only marginal improvements. The enhanced conversion can be attributed to higher sweep gas flow rates creating a stronger driving force for H_2_O removal, which shifts the reaction equilibrium toward product formation. Based on these findings, increasing the sweep gas flow rate by 5–15 times the reference value was determined to be optimal for maximizing CO_2_ conversion efficiency.

Based on the findings, the system configuration was updated to maximize efficiency. Table 9 presents a comparison between the initial and updated reaction conditions. As shown in Figure 12, the updated configuration achieved significantly better performance, with conversion rates approaching unity and maximum water removal efficiency. The improved reaction conditions led to enhanced conversion at lower temperatures, demonstrating a substantial improvement in RWGS reaction efficiency. This optimization resulted in a nearly complete conversion (0.99) with optimal water removal (0.99), compared to the initial configuration which achieved only moderate conversion (0.61) and water removal (0.53).

As shown in Figure 12, the conversion approaches 0.99 instead of 1, confirming that the system is still governed by practical constraints in water removal through membrane permeation.

### 3.3. System Development for RWGS Reaction in MPBR (ZSM-5 Membrane)

#### 3.3.1. Experiment vs. Simulation

Figure 13 presents a comparison between simulation results and experimental data for outlet gas composition across various temperatures. The results demonstrate excellent agreement between our model predictions and the experimental measurements reported by Sakai et al. This strong correlation validates the accuracy and reliability of our developed simulation model, confirming its ability to effectively represent the actual behavior of the RWGS reaction system under different operating conditions. Figure 14 also shows a comparison between experimental and simulated conversion. The simulation results demonstrate excellent agreement with experimental data, closely matching the conversion values observed in laboratory tests. This strong correlation, combined with the previously observed alignment in gas composition profiles on both sides of the reactor, provides strong evidence that our model accurately represents the actual experimental system.

From this point onwards, the simulation will use actual catalyst and membrane data to replicate the real system, necessitating system validity testing. To ensure an accurate simulation of the real system based on actual catalyst and membrane data, it is crucial to validate whether a one-dimensional (1D) model is suitable for describing the reactor system. The validity of the 1D approach is determined by evaluating several key criteria.

First, the dominance of axial flow is confirmed by the significantly high Péclet number (3338.46) by Equation (28), which indicates that axial convection far exceeds diffusion. Additionally, radial diffusion is negligible due to the small effective diffusion coefficient (4.345 × 10^−4^ cm^2^/s), ensuring that mass transfer in the radial direction does not impact the overall performance.

The system also satisfies the laminar or plug flow assumption, as evidenced by the low superficial velocity (1.45 mm/s) in the packed bed. Furthermore, axial reaction homogeneity is achieved through the combination of a high Péclet number and a short reactor length (0.09 m), ensuring uniform reaction conditions along the reactor’s axial direction. Moreover, radial effects are minimal, as lateral diffusion is small and advection remains the dominant transport mechanism in this system.

A critical evaluation based on the Péclet number criterion by Lundin et al. further supports the use of a 1D model (Equations (29)–(35)) [67]. At 609 K, the critical Péclet number (Pe_crit_) is calculated as 9.9493 × 10^−4^, which is much lower than the actual Péclet number (3338.46). Since the system satisfies the condition Pe > Pe_crit_, the 1D approach is justified, reducing computational complexity while ensuring accurate results.

Mathematically, the Péclet number (Pe) is determined in Equation (28). The effective diffusion coefficient is determined using Equations (29)–(35), accounting for catalyst particle size and inner reactor diameter. The critical Péclet number (Pe_crit_) is derived from an empirical equation, ensuring that the correct transport regime is identified. Data that summarize the determination of the use of the 1D model can be seen in Table 10.

In conclusion, all required criteria for applying a 1D reactor model are met. Given that Pe ≫ Pe_crit_, the system can be accurately modeled using a 1D approach, providing a computationally efficient yet reliable representation of the reactor behavior. However, if Pe ≤ Pe_crit_, a 2D model would be necessary to capture the additional transport effects and improve accuracy.

#### 3.3.2. Analysis of Parameter Changes in MPBR with ZSM-5 Membrane

Figure 15 presents the calculated relationship between water removal and CO_2_ conversion across different temperatures. These calculations serve as a validation benchmark, establishing the theoretical maximum conversion limits for varying degrees of water removal. By comparing simulation results against these theoretical limits, we can verify that the model maintains physical accuracy and does not predict conversions beyond thermodynamic equilibrium constraints.

Figure 16 demonstrates the temperature-dependent behavior of the ZSM-5 membrane, where increasing temperature has opposing effects on reactants and products. As temperature rises, the membrane shows enhanced permeation of reactants (CO_2_ and H_2_) while simultaneously reducing the permeation of products (CO and H_2_O). This unbalanced permeation prevents the system from reaching reaction equilibrium because reactants continuously exit the system rather than participating in the reaction, effectively inhibiting the desired shift toward product formation.

Since the reactants continue to exit the reaction system, we compared the efficiency of the system if only H_2_O could pass through the membrane (called an ideal H_2_O membrane). Figure 17 shows a comparison between the ideal H_2_O membrane, which only permeates water, and the ZSM-5 membrane used today.

As expected, an ideal membrane that exclusively permeates water results in higher reaction conversions under identical conditions. However, there is an interesting trade-off: although the ideal membrane has higher selectivity, the overall amount of water eliminated is less compared to the ZSM-5 membrane.

This phenomenon occurs because the lower selectivity of the ZSM-5 membrane allows the removal of water not only directly but also indirectly through the absorption of other components. These findings reveal an important balance between membrane selectivity and molecular transport capacity in RWGS reactor design.

The simulation findings of the ZSM-5 membrane system (Figure 18) revealed that at high temperatures, the membrane reactor has a lower conversion compared to the traditional PBR due to the higher permeability of the reactants (CO_2_ and H_2_). While the experimental study of Sakai et al. showed a conversion rate of 2–3% above equilibrium at 241–336 °C, our simulations showed the same at 250–400 °C, which allows us to conclude that the simulation model is reliable as it accurately predicts the general behavior and trends observed in the experiments.

Increasing the sweep gas rate from 5 to 200 mL/min significantly enhanced reaction conversion in the temperature range of 200–600 °C, pushing the reaction well above PBR equilibrium (Figure 19). This improvement occurs because higher sweep gas rates facilitate more efficient water removal from the system. However, at higher temperatures, an interesting phenomenon emerges: ZSM5 systems with larger sweep gas rates show lower conversion rates. This counter-intuitive result stems from two factors: decreased water permeation efficiency and increased reactant permeation. The higher sweep gas rate not only reduces water removal capability but also promotes unwanted permeation of reactants, ultimately depleting the reactant concentration in the system and diminishing overall conversion efficiency.

In a system with a 200 mL/min sweep gas flow, we investigated the effect of selectively deactivating membrane permeation for different components (CO_2_, H_2_, and CO) (Figure 20a). When CO_2_ permeation was deactivated, the conversion increased significantly since CO_2_ was the primary reactant. Deactivating H_2_ permeation also improved conversion, although to a lesser extent, as the excess H_2_ in the system ensures sufficient reactant availability even with some H_2_ loss through permeation. Deactivating CO permeation showed minimal impact due to its inherently low permeation rate. These findings indicate that developing membranes with high CO_2_ resistance is crucial for optimizing RWGS reaction conversion.

Varying the total feed gas flow rate between 5, 12, and 19 mL/min showed no impact on the maximum achievable conversion, which remained constant at 0.8 across all conditions (Figure 20b). However, the system reached maximum conversion more rapidly at lower feed flow rates. Based on these findings, a feed flow rate of 5 mL/min was selected for subsequent experiments as it provided the most efficient path to reaching maximum conversion.

In this optimization study, we also investigated the effect of modifying H_2_O permeation while keeping other component permeation rates constant (Figure 20c). The results show that a 10-fold increase in H_2_O permeance is optimal for the system’s performance. Further increases beyond this point provide minimal benefits because at 10 times the original permeance (*J_o_* times 10), nearly all CO_2_ is consumed from the reaction side. While increasing H_2_O permeance does indirectly increase CO_2_ permeation through partial pressure changes, the conversion rate plateaus once CO_2_ becomes depleted on the reaction side. Therefore, designing a membrane with 10 times higher H_2_O permeability represents the optimal target for this system.

Multiple optimization efforts were implemented to enhance the system’s performance from its initial operating conditions. The key modifications included reducing the total feed flow rate from 12 to 5 mL/min, substantially increasing the sweep gas flow from 5 to 200 mL/min, and elevating the operating temperature from 336 °C to 350 °C. These adjustments resulted in significantly improved system performance, with notably higher conversion rates compared to the initial configuration. The optimization effects are evident in the changes observed in both flow rate patterns and partial pressure profiles across all components in the system (Figure 21).

#### 3.3.3. Modifying Membrane Permeability to CO_2_ for RWGS Reaction in MPBR with ZSM-5 Membrane

A comparison between experimental and simulation data reveals interesting insights about separation factors (SFs) in the RWGS system. While experimental H_2_O separation factors for the complete RWGS reaction are not directly available, binary mixture test data from Sakai et al. provide a valuable reference point. The simulation model successfully calculated H_2_O separation factors for each component in the RWGS system, showing lower values compared to binary mixture tests (Figure 22). This reduction in separation factors is logical, as the presence of multiple components in the RWGS reaction system impedes effective H_2_O permeation through the membrane. Despite these lower separation factors, the model’s validity is confirmed by the close alignment between simulated and experimental results in terms of conversion rates and component composition. This demonstrates that the developed simulation model effectively captures the complex behavior of the RWGS membrane reactor system.

The effect of CO_2_ permeance modification on membrane performance varies across different temperature ranges, as can be seen in Figure 23a. Below 300 °C (green region), changes in CO_2_ permeance have minimal impact as the membrane effectively manages water permeation with naturally low CO_2_ transfer. In the intermediate temperature range (300–700 °C or pink region), reducing CO_2_ permeance to one-tenth of its initial value significantly improves conversion. The optimal improvement occurs with a 10-fold reduction in CO_2_ permeance, while further reductions to 25 or 50 times show only marginal benefits, approaching the performance of a completely CO_2_-impermeable membrane. At higher temperatures (higher than 700 °C or yellow region), additional reductions in CO_2_ permeance become less practical due to diminishing returns and operational challenges. Therefore, a 10-fold reduction in CO_2_ permeance represents the most efficient optimization strategy, balancing improved conversion with practical implementation.

The optimization of membrane performance in the RWGS reaction system provides valuable insights into separation factors and system efficiency (Figure 23b). Under specific operating conditions (600 °C, 1 atm, H_2_O/CO_2_ = 3, 12 mL/min feed flow, 200 mL/min sweep flow), reducing CO_2_ permeance by a factor of 10 increased the H_2_O/CO_2_ separation factor from 15.35 to 148.12, enhancing conversion rates. Although permeation rates for other components remained constant, their separation factors changed due to composition shifts in the permeate side. Measuring separation factors in actual RWGS systems is experimentally challenging, with binary mixture tests typically showing higher separation factors than complete RWGS systems due to their simpler conditions. Future membrane development should focus on achieving separation factors at least equal to those observed in RWGS simulations, with the goal of matching or exceeding binary mixture test values. Controlling pore size offers a promising strategy for managing CO_2_ permeance and achieving these targets, potentially improving overall RWGS reaction system efficiency.

#### 3.3.4. Reactor Design for RWGS Reaction in MPBR with ZSM-5 Membrane

Previous studies revealed that excessive CO_2_ permeation at the reactor’s entrance significantly reduced system efficiency, as this key reactant was being removed before it could effectively participate in the reaction. To address this limitation, a new reactor design called Half-MPBR was developed. This design features a shortened membrane that begins partway through the reactor rather than at the entrance. By positioning the membrane further along the packed bed, the reactants can initially undergo the RWGS reaction without premature removal, leading to improved reaction efficiency. This strategic placement of the membrane prevents early CO_2_ loss while maintaining the benefits of product removal in the latter portion of the reactor. Not only that, a shorter membrane means lower membrane manufacturing costs, so it will also be effective in terms of financing.

The selection of the membrane starting point in the Half-Membrane Packed Bed Reactor (Half-MPBR) design has a significant influence on the conversion profile throughout the reactor as well as the final conversion at the end of the reactor. From the simulation results, the variation in the membrane starting point from 0.01 m to 0.09 m results in different conversion trends, which shows how important the membrane placement is in optimizing the reactor performance.

From the conversion profile along the length of the reactor (Figure 24a), if the membrane is placed too early (e.g., at 0.01–0.02 m), CO_2_ as the main reactant starts to remain too heavily permeated before it has a chance to react optimally, causing a slower increase in conversion. Conversely, if the membrane is placed further into the reactor (0.03–0.07 m), the conversion increases faster in the early part before eventually reaching stability, indicating a better balance between reaction and separation. However, if the membrane is placed too late or even not used at all (0.08–0.09 m, as in a conventional PBR reactor), the conversion remains low and tends to be limited by the equilibrium of the reaction, meaning the system loses the benefit of removing the reaction products to drive the equilibrium shift.

In theory, this result can be explained through the nature of the Reverse Water Gas Shift (RWGS) reaction which is an equilibrium-limited reaction. In the absence of product separation, conversion will remain limited according to thermodynamic equilibrium. If the membrane is placed too early, CO_2_ separation limits the ongoing reaction. However, if the membrane is placed too late, the system loses the opportunity to utilize product separation as a further reaction driver. Therefore, the best approach is to place the membrane after the reaction start zone (around 0.02–0.06 m), so that the reactants have sufficient time to react before the products are separated, which ultimately increases the conversion efficiency significantly.

From the results of conversion at the end of the reactor (Figure 24b), it can be observed that high conversion (>0.94) is obtained when the membrane starts working in the range of 0.03–0.07 m, where the highest conversion is obtained when the membrane is placed at approximately half of the length of the packed bed, which is 0.05 m, but the results are not much different from the shorter membrane (0.06 or 0.07). Based on these results, it can be concluded that in the half-membrane system, the membrane will be effective if it starts from 70 to 80% of the total length of the catalyst bed. If the length of the catalyst bed is 0.09, then the membrane will be optimal if it starts from 0.063 to 0.072 from the starting point of the catalyst bed. In this way, using only 20–30% of the membrane length of the total catalyst bed length, we can improve the system and reduce membrane production costs in one step. These results show that the selection of the membrane starting point is a crucial aspect of membrane reactor design, especially to improve the conversion efficiency in RWGS-based systems.

Furthermore, by using the maximum point (membrane starting from point 0.05), the Half-Membrane Packed Bed Reactor (Half-MPBR) design is compared with the previous usual design. The Half-Membrane Packed Bed Reactor (Half-MPBR) demonstrated superior performance compared to both conventional PBR and standard MPBR configurations (Figure 25). When combined with reduced CO_2_ permeability (JCO_2_/10 model), the system’s efficiency is improved even further. At elevated temperatures of 650 °C, both the Half-MPBR and JCO_2_/10 MPBR configurations achieved near-maximum conversion rates. The most effective configuration proved to be the Half-MPBR with JCO_2_/10 modification, highlighting that minimizing early CO_2_–membrane interaction at the reactor’s entrance significantly enhances overall system performance.

## 4. Conclusions

This study optimized the Reverse Water Gas Shift (RWGS) reaction in a membrane reactor to enhance CO_2_ conversion to CO and H_2_O, using a 0.5 wt% Ru-Cu/ZnO/Al_2_O_3_ catalyst, ZSM5 membrane, and data-driven simulations. The model’s reliability was confirmed through experimental validation. The results demonstrated that a Membrane Packed Bed Reactor (MPBR) with an ideal H_2_O-selective membrane out-performed a conventional Packed Bed Reactor (PBR) in CO_2_ conversion, surpassing thermodynamic equilibrium limitations. A comparison of MPBR-In and MPBR-Out configurations showed minimal performance differences for small-scale reactors. Analysis of co-current and counter-current flow in MPBR revealed that co-current flow achieved higher conversion at low W/F ratios and crossed the equilibrium limit more quickly, while counter-current flow excelled at high W/F ratios. Although the ZSM-5 membrane performed below the ideal membrane due to the permeation of the reactants, its ability showed potential for broader applications with advanced membrane designs. Half-membrane reactor design is the key to overcoming this problem and has been shown to improve system efficiency. In half-membrane systems, membranes placed at 70–80% of the total catalyst bed length show preferable efficiency. This study highlights that low CO_2_ permeation resistance in membrane design, the development of membranes with better water permeation, and optimized operational parameters are crucial for effectively applying RWGS in carbon emission reduction.

## Figures and Tables

**Figure 1 membranes-15-00107-f001:**
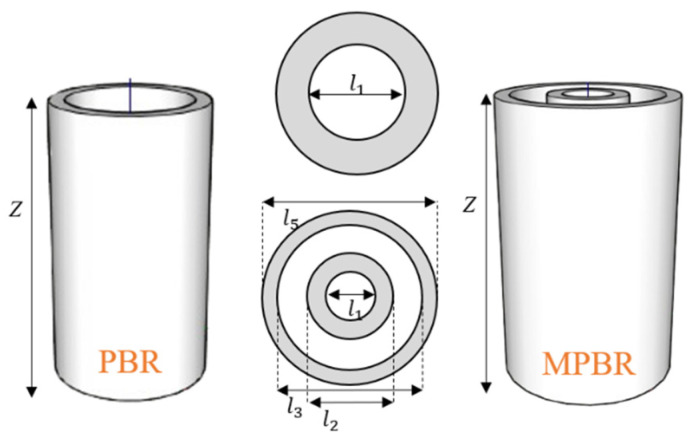
Comparison of the PBR (**left**) and MPBR (**right**) systems.

**Figure 2 membranes-15-00107-f002:**
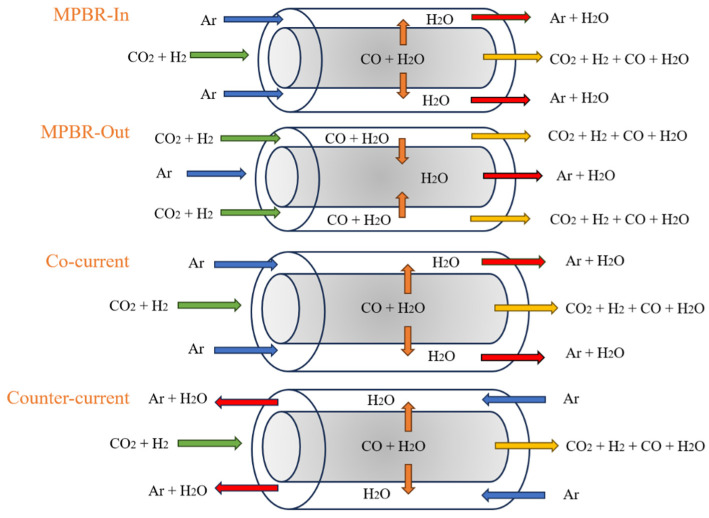
Configuration tests on the MPBR system.

**Figure 3 membranes-15-00107-f003:**
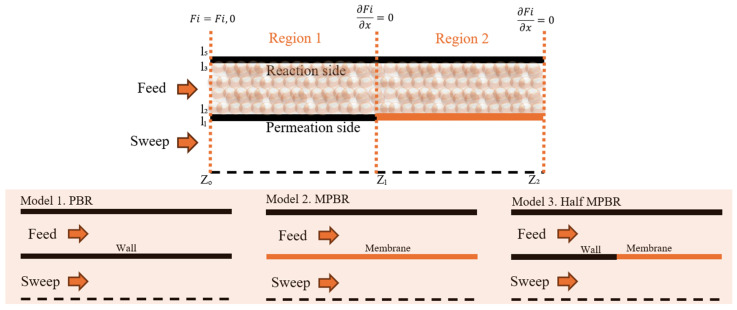
Reactor design.

**Figure 4 membranes-15-00107-f004:**
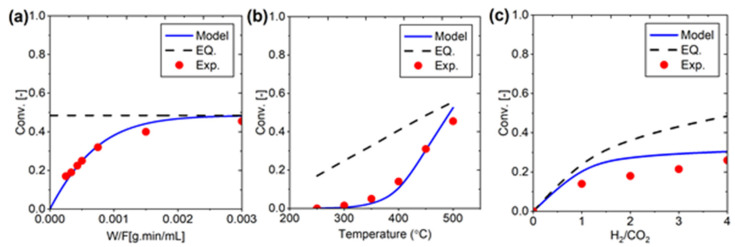
Experiment versus simulation for PBR system development: (**a**) Conversion as a function of W/F, (**b**) Conversion as a function of temperature, and (**c**) Conversion as a function of H₂/CO₂ ratio.

**Figure 5 membranes-15-00107-f005:**
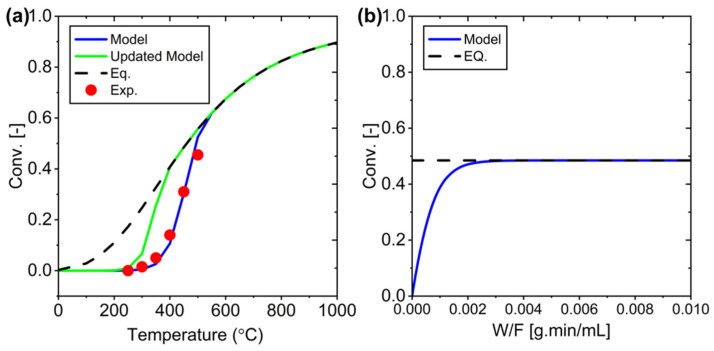
Optimal conditions for the PBR system: (**a**) Conversion vs. temperature, and (**b**) Conversion vs. W/F at 400 °C (the optimal temperature).

**Figure 6 membranes-15-00107-f006:**
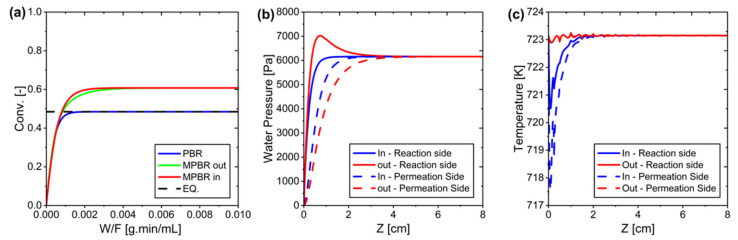
(**a**) CO_2_ conversion profile in different reactor configurations. (**b**) Water pressure distribution along the reactor for the reaction and permeation side. (**c**) Temperature profile along the reactor, where minor oscillations at the inlet region are observed due to localized reaction effects before reaching thermal stability.

**Figure 7 membranes-15-00107-f007:**
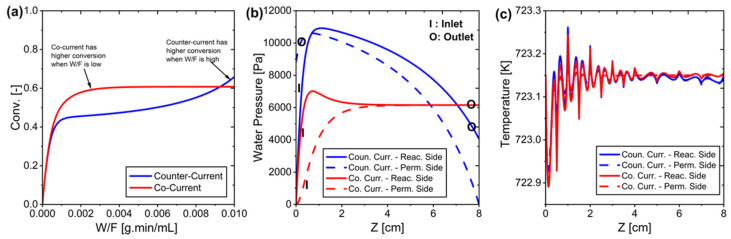
Results of co-current and counter-current arrangements: (**a**) Conversion vs. W/F, (**b**) Water pressure on the reaction and permeation sides along the system, and (**c**) Temperature distribution along the system.

**Figure 8 membranes-15-00107-f008:**
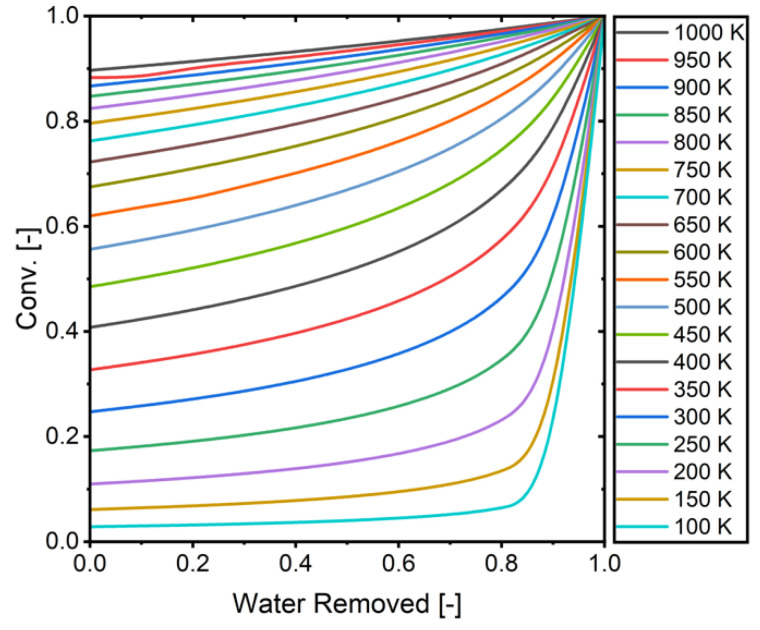
The relationship between removed water and conversion (H_2_/CO_2_ = 4).

**Figure 9 membranes-15-00107-f009:**
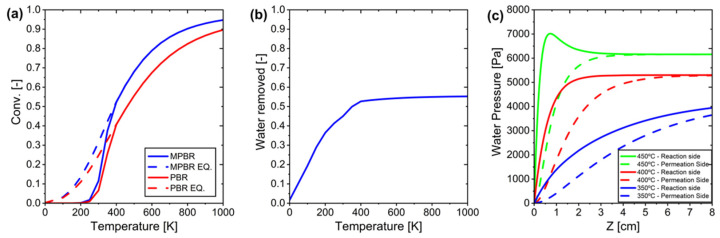
System performance with the ideal membrane at various temperatures: (**a**) conversion vs temperature, (**b**) only water removed profile at different temperature, and (**c**) water pressure profile at reaction and permeation side at different temperatures.

**Figure 10 membranes-15-00107-f010:**
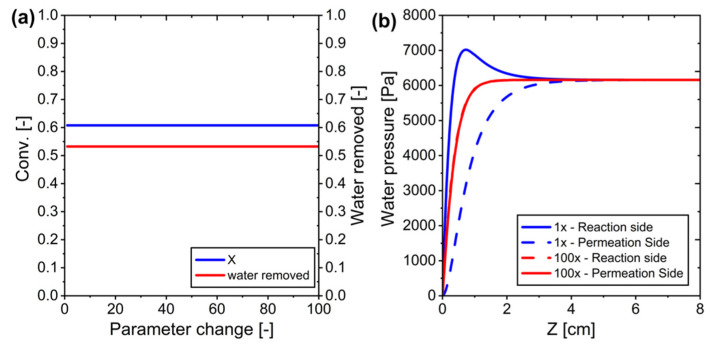
Modifying water permeance in ideal membrane systems: (**a**) Conversion versus changes in parameters, and (**b**) Water pressure across the system for parameter *J* is equal to 1 and when multiplied by 100.

**Figure 11 membranes-15-00107-f011:**
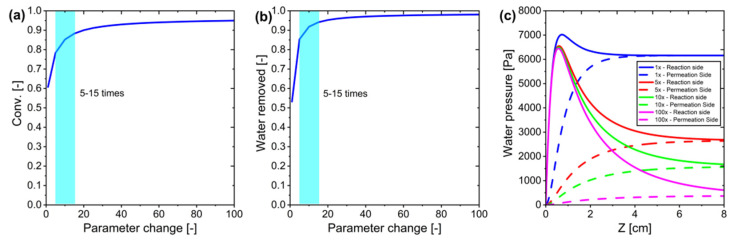
Modifying the permeate side sweep gas flow rate in ideal membrane systems: (**a**) Conversion as a function of parameter change, (**b**) Water pressure profile as a function of parameter change, and (**c**) Water pressure profile on the reaction and permeation sides along the system for parameter changes of 1×, 5×, 10×, and 100×.

**Figure 12 membranes-15-00107-f012:**
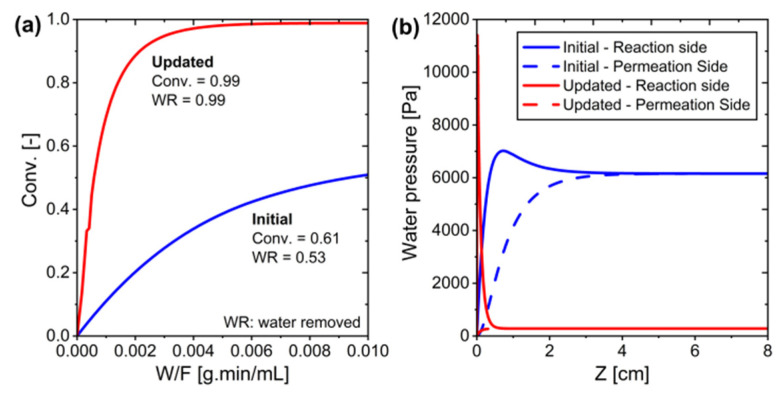
The initial and updated conditions in the MPBR ideal membrane system: (**a**) Conversion versus W/F, and (**b**) Water pressure along the system.

**Figure 13 membranes-15-00107-f013:**
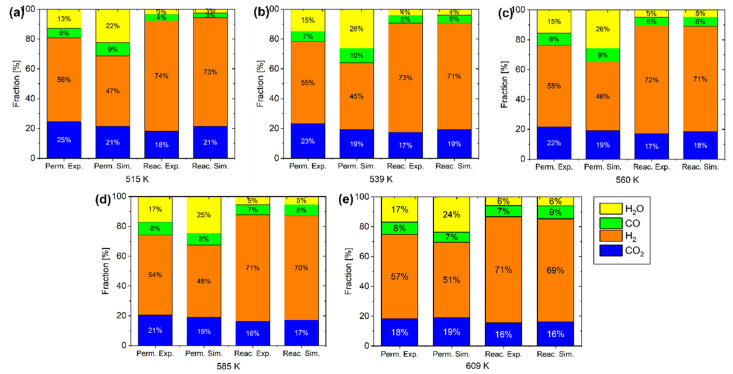
Comparison of gas composition at the outlet (experimental vs. simulation) at various temperatures: (**a**) 515 K, (**b**) 539 K, (**c**) 560 K, (**d**) 585 K, and (**e**) 609 K.

**Figure 14 membranes-15-00107-f014:**
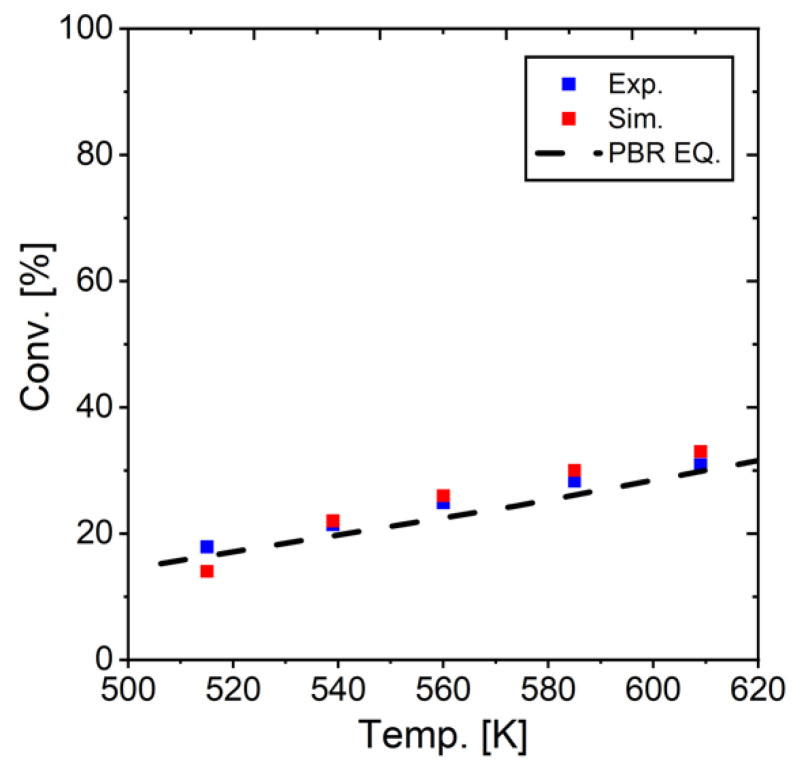
Comparison of simulation and experimental results at different temperatures.

**Figure 15 membranes-15-00107-f015:**
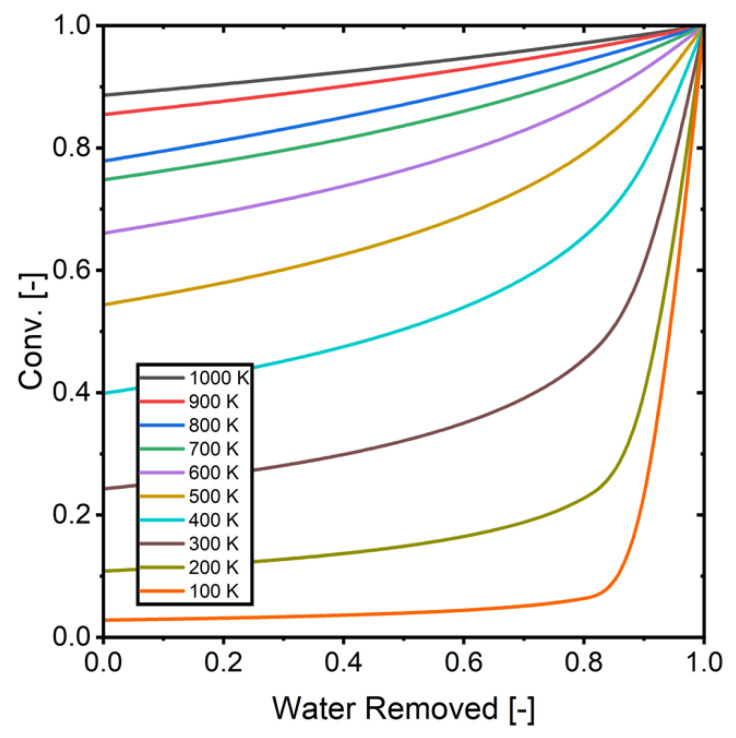
The relationship between removed water and conversion (H_2_/CO_2_ = 3).

**Figure 16 membranes-15-00107-f016:**
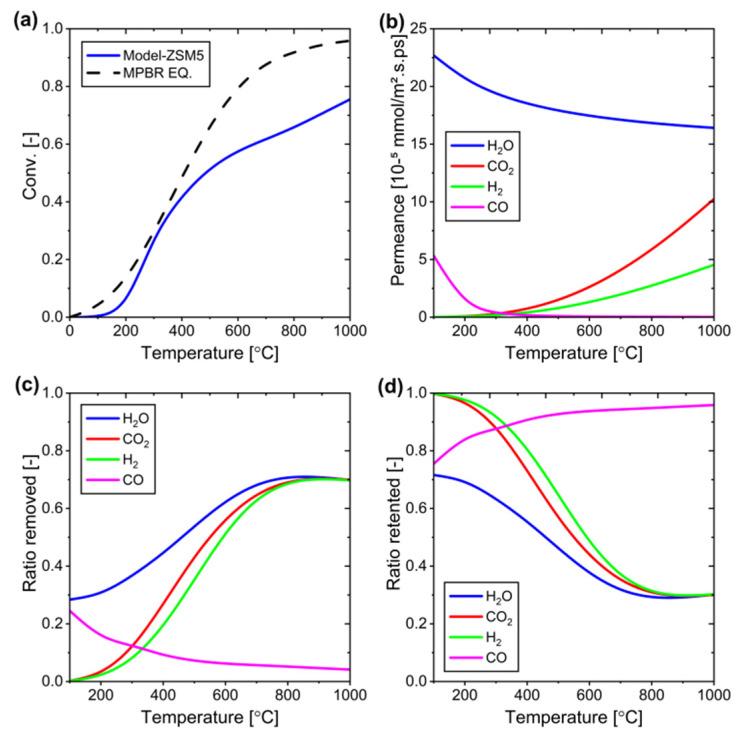
Performance of the ZSM-5 membrane at different temperatures: (**a**) Conversion versus temperature, (**b**) Changes in permeance of each component versus temperature, (**c**) Component removal (permeation) ratio at different temperatures, and (**d**) Component retention ratio at different temperatures.

**Figure 17 membranes-15-00107-f017:**
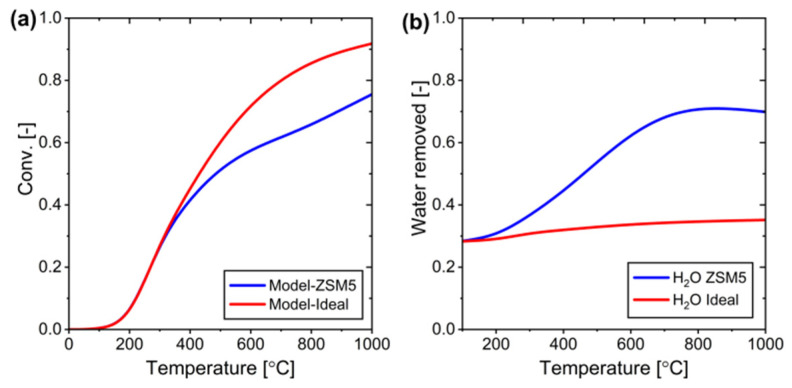
Performance comparison between ZSM-5 membrane and ideal membrane at different temperatures: (**a**) conversion as a function of temperature, (**b**) water rejection as a function of temperature.

**Figure 18 membranes-15-00107-f018:**
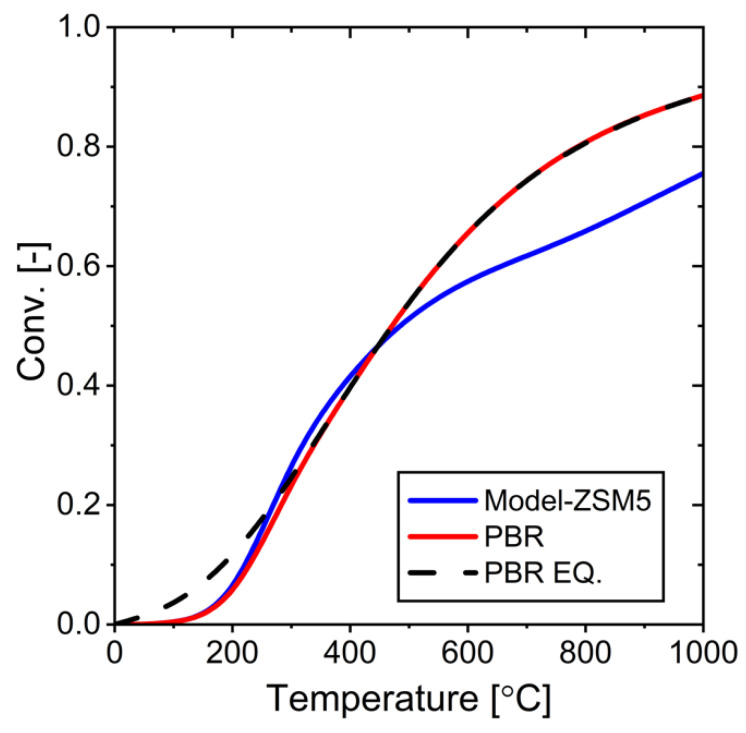
Performance comparison of PBR vs. MPBR at different temperatures.

**Figure 19 membranes-15-00107-f019:**
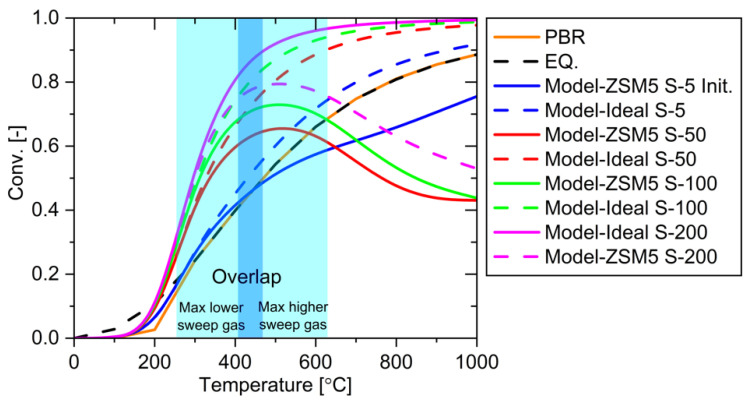
System performance using various temperatures and sweep gases.

**Figure 20 membranes-15-00107-f020:**
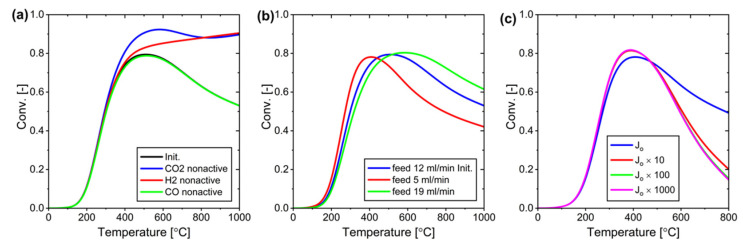
(**a**) Influence of deactivating components on the ZSM-5 System, (**b**) Impact of varying feed gas flow rates on conversion, and (**c**) Effect of H_2_O permeance on conversion efficiency.

**Figure 21 membranes-15-00107-f021:**
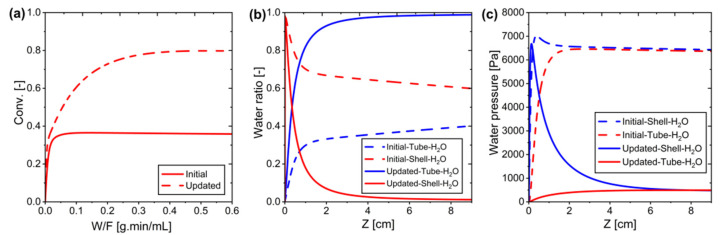
Comparison of initial and updated systems: (**a**) Conversion vs. W/F, (**b**) Water ratio on the tube side and shell side along the system, and (**c**) Water pressure along the system.

**Figure 22 membranes-15-00107-f022:**
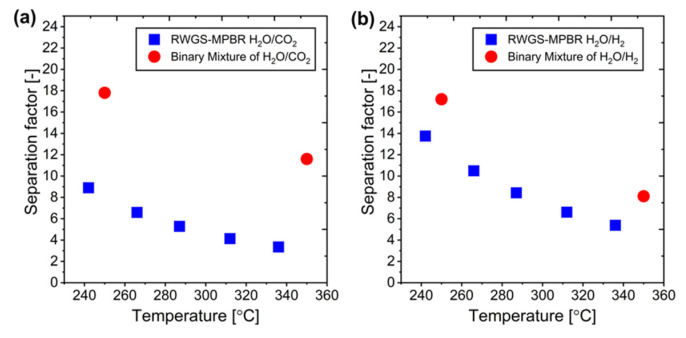
Separation factors: MPBR vs. binary mixture: (**a**) H_2_O versus CO_2_, and (**b**) H_2_O versus H_2_.

**Figure 23 membranes-15-00107-f023:**
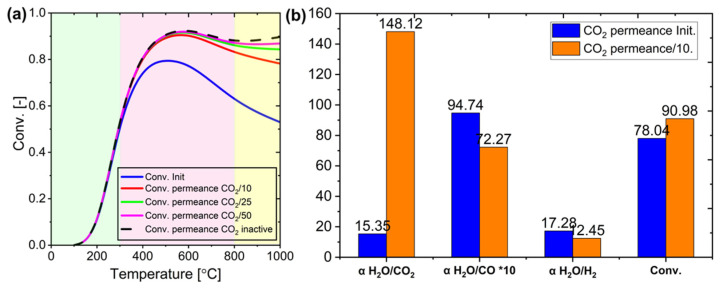
(**a**) Impact of membrane CO_2_ permeance change on conversion at various temperatures, and (**b**) binary mixture separation factor at 600 °C under initial conditions (Init.) and when the membrane retains CO_2_ at 1/10 of its initial permeance.

**Figure 24 membranes-15-00107-f024:**
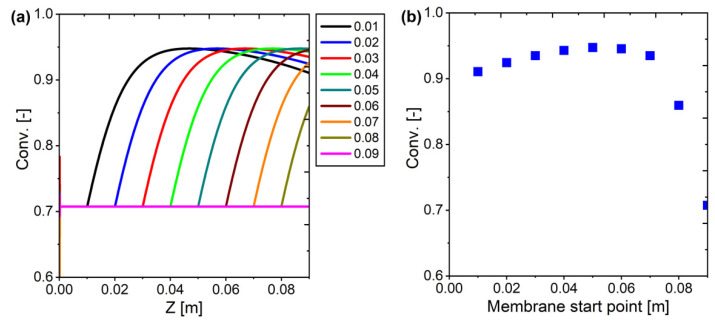
Membrane starting point selection: (**a**) profile across the reactor; (**b**) conversion at the outlet.

**Figure 25 membranes-15-00107-f025:**
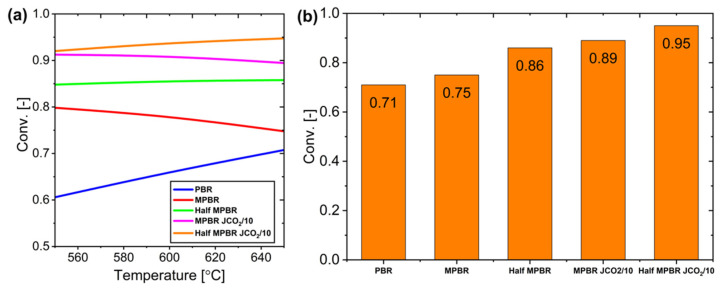
The results of the proposed system: (**a**) Conversion at various temperatures, and (**b**) Conversion measured at the outlet at 650 °C for different configurations.

**Table 1 membranes-15-00107-t001:** Summary of catalyst properties and reaction conditions.

Catalyst Properties [33]	
Activation energy [kJ/mol]	96.8
Frequency factor [mol/kg_-cat_.s.bar] @643K	450
Adsorption constant of CO_2_ [/bar]	0
Adsorption constant of CO [/bar] @648K	0.6763
Adsorption constant of H_2_O [/bar] @823K	0.0042
Adsorption enthalpy change in CO [kJ/mol]	−70.91
Adsorption enthalpy change in H_2_O [kJ/mol]	88.68
**Reaction Conditions** [33]	
Catalyst weight [g]	0.5
Packed bed density [g/cm^3^]	5
Packed bed length [cm]	1.3

**Table 2 membranes-15-00107-t002:** Reaction conditions for each condition in the experiment and simulation in PBR.

W/F		H_2_/CO_2_		Temp.	
Pressure [psi]	45	Pressure [psi]	45	Pressure [psi]	45
Temperature [°C]	450	Temperature [°C]	450	H_2_/CO_2_	4
H_2_/CO_2_	4	GHSV [mL/g·h]	90,000	GHSV [mL/g·h]	90,000

**Table 3 membranes-15-00107-t003:** Physical property value data for simulation.

	CO_2_	H_2_	CO	H_2_O	Ar
Δ*H_f,298K_* [kJ/mol]	−3.94 × 10^2^	0.00 × 10^0^	−1.11 × 10^2^	−2.42 × 10^2^	0.00 × 10^0^
*Sº* [J/mol.K]	2.14 × 10^2^	1.31 × 10^2^	1.98 × 10^2^	1.89 × 10^2^	0.00 × 10^0^
*ai* [J/mol/K]	2.74 × 10	2.54 × 10^1^	2.96 × 10^1^	3.39 × 10^1^	2.08 × 10^1^
*bi* [J/mol/K^2^]	4.23 × 10^−2^	2.02 × 10^−2^	-6.58 × 10^−3^	−8.42 × 10^−3^	0.00 × 10^0^
*ci* [J/mol/K^3^]	−1.96 × 10^−5^	−3.85 × 10^−5^	2.01 × 10^−5^	2.99 × 10^−5^	0.00 × 10^0^
*di* [J/mol/K^4^]	4.00 × 10^−9^	3.19 × 10^−8^	−1.22 × 10^−8^	−1.78 × 10^−8^	0.00 × 10^0^
*ei* [J/mol/K^5^]	−2.99 × 10^−13^	−8.76 × 10^−12^	2.26 × 10^−12^	3.69 × 10^−12^	0.00 × 10^0^
*M*_w_ [g/mol]	4.40 × 10	2.02 × 10^0^	2.80 × 10^1^	1.80 × 10^1^	3.99 × 10^1^
*K* [Pas/K^0.5^]	1.37 × 10^−5^	8.41 × 10^−6^	1.65 × 10^−5^	1.71 × 10^−5^	-
*C* [K]	2.40 × 10^2^	7.20 × 10^1^	1.18 × 10^2^	5.63 × 10^2^	1.51 × 10^2^
*ρ0,i* [kg/m^3^]	1.98 × 10^0^	8.99 × 10^−2^	1.25 × 10^0^	1.00 × 10^3^	1.78 × 10^0^

**Table 4 membranes-15-00107-t004:** Reaction conditions for ideal MPBR membrane.

Reaction Conditions	
Total initial feed volumetric flow rate [mL/min]	50
H_2_/CO_2_	4
Catalyst weight [mg]	500
Packed bed density [g/m^3^]	97,261
Total initial sweep volumetric flow rate [mL/min]	50
Pressure [atm]	1
Temperature [K]	723.15
**Membrane Properties**	
Activation energy [kJ/mol]	14.05
Frequency factor [mol/m^2^·s·Pa]	7.57 × 10⁻⁶

**Table 5 membranes-15-00107-t005:** Reactor dimensions.

Reactor Type	PBR	MPBR-In, Co-Current, and Counter-Current	MPBR-Out
*Z* [m]	8.00 × 10^−2^	8.00 × 10^−2^	8.00 × 10^−2^
*l*_1_ [m]	1.79 × 10^−2^	1.79 × 10^−2^	7.00 × 10^−3^
*l*_2_ [m]		2.19 × 10^−2^	1.10 × 10^−2^
*l*_3_ [m]		2.30 × 10^−2^	2.10 × 10^−2^
*l*_5_ [m]		2.50 × 10^−2^	2.50 × 10^−2^
*S*_1_ [m^2^]	2.01 × 10^−5^	2.01 × 10^−5^	3.08 × 10^−6^
*S*_2_ [m^2^]		3.13 × 10^−6^	2.01 × 10^−5^
*V*_1_ [m^3^]	2.50 × 10^−5^	2.51 × 10^−4^	1.60 × 10^−5^
*V*_2_ [m^3^]		1.60 × 10^−5^	2.51 × 10^−4^

**Table 6 membranes-15-00107-t006:** The experimental conditions and dimensions used in the ZSM5 membrane system.

Experiment Conditions			System Illustration
Total pressure [Pa]		101,325	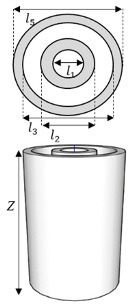
Total feed shell side [mL/min]		12	
Total feed tube side [mL/min]		5	
H_2_/CO_2_ ratio		3:01	
Catalyst weight [g]		3.5	
Density of catalyst bed [g/m^3^]		4.5	
Dimension			
Z [m]		9 × 10^−2^	
l1 [m]		7 × 10^−3^	
l2 [m]		1 × 10^−2^	
l3 [m]		1.2 × 10^−2^	
l5 [m]		1.5 × 10^−2^	
Membrane properties			
	$J_{0}$ [mmol/m^2^ s Pa]	$E_{a}$ [mmol/m^3^ Pa]	
CO_2_	2.1 × 10^−3^	32	
H_2_	7.08 × 10^−4^	29.1	
CO	1.91 × 10^−8^	−24.6	
H_2_O	1.43 × 10^−4^	−1.42	

**Table 7 membranes-15-00107-t007:** Reaction conditions for reactor design.

Parameter [Unit]	Value
Pressure [atm]	1
H_2_O/CO_2_	3
Feed [mL/min]	12
Sweep [mL/min]	200
$\rho_{cat}$ [g/m^3^]	4,500,000
Z_o_ [m]	0
Zₗ [m]	0.5
Z_2_ [m]	0.9
lₗ [m]	0.007
l_2_ [m]	0.01
l_3_ [m]	0.012
l_5_ [m]	0.015

**Table 8 membranes-15-00107-t008:** Changes from the initial reaction to the updated conditions in the PBR system.

	Initial Configuration—From Experimental Conditions	Updated Configuration—Optimized Process Variables
Pressure [Pa]	310,264	101,325
Total volumetric flow rate [mL/min]	750	50
H_2_/CO_2_	4	4
Temperature at equilibrium is reached [°C]	~500	~450

**Table 9 membranes-15-00107-t009:** Changes from the initial to the updated configuration in the MPBR ideal membrane system.

	Initial Configuration	Updated Configuration
Shell side		
Total initial volumetric flow rate [mL/min]	50	5
H_2_/CO_2_	4	4
Catalyst weight [mg]	500	500
Packed bed density [g/m^3^]	97,262	97,262
Tube side		
Total initial volumetric flow rate [mL/min]	50	350
Reactor general setting		
Pressure [atm]	1	1
Temperature [K]	723.15	823.15
Membrane properties [23]		
Activation energy [kJ/mol]	14.1	14.1
Frequency factor [mol/m^2^ s Pa]	7.57 × 10^−6^	7.57 × 10^−6^

**Table 10 membranes-15-00107-t010:** Criteria for applying the 1D model in the system (the verification of the system’s validity).

Criteria	Met/Not Met	Reason
Dominance of axial flow	✅	The Péclet number is very high (3338.46), indicating that axial flow dominates over diffusion.
Radial diffusion negligible	✅	The effective diffusion coefficient is small (0.0004345 cm^2^/s), and a high Péclet ensures radial diffusion is insignificant.
Laminar or plug flow	✅	Low flow velocity (1.45 mm/s) supports the plug flow assumption in the packed bed.
Axial reaction homogeneity	✅	High Péclet number and short reactor length (0.09 m) ensure axial reaction homogeneity.
Radial effects negligible	✅	Small lateral diffusion and dominance of advection minimize radial effects in this system.
Péclet number criterion [67]	✅	The calculated Pecrit is lower than the Péclet number: @609K Pe_crit_ (0.099493) vs. Pe (3338.460)

## Data Availability

The original contributions presented in this study are included in the article. Further inquiries can be directed to the corresponding author(s).

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
