# Peer review of "Optimization of Carbon Dioxide Utilization: Simulation-Based Analysis of Reverse Water Gas Shift Membrane Reactors"

_membranes, 2025, doi:10.3390/membranes15040107_

Round 1
Reviewer 1 Report
Comments and Suggestions for Authors
This paper presents a simulation and optimization study of the RWGS membrane reactor system, which integrates a ZSM-5 water-permeated membrane with a Ru-Cu/ZnO/Al2O3 catalyst. The study provides valuable insights, particularly in the optimization of key parameters. The introduction of the Half MPBR as a new configuration for the RWGS membrane reactor is an interesting and novel concept. However, the manuscript requires several revisions before it can be considered for publication.
- Page 12 Line 318: The term "hydrogen pressure" should be replaced with "water pressure" for clarity, as the context suggests that water pressure is being discussed.
- Page 15 Line 405: The reference to "Figure 13" should be corrected to "Figure 14." Additionally, it would be beneficial that Figure 14 includes the calculated COâ‚‚ conversion limit, which is obtained from the thermodynamic equilibrium limitation in the PBR for better clarity.
- The Peclet number plays a crucial role in the performance of RWGS membrane reactors. Please provide a more detailed explanation of the Peclet number and its relevance to the system under study.
- There are a significant number of equations presented in the manuscript, and they may be confusing to readers. It would be helpful to briefly explain each equation and its purpose within the context of the study to improve readability and understanding.
- The concept of the Half MPBR is quite interesting and innovative. I suggest further elaborating on its advantages and potential applications in the RWGS process, as this could greatly enhance the impact of the study.
Author Response
For research article
Response to Reviewer 1 Comments
|
||
1. Summary |
|
|
Thank you very much for taking the time to review this manuscript. We sincerely appreciate your valuable feedback and constructive comments. Please find our detailed responses below, along with the corresponding revisions and corrections, which have been highlighted in the re-submitted files.
We recognize the importance of addressing each point raised, and where necessary, we have provided clarifications or alternative perspectives while maintaining a respectful and scientific discourse.
Once again, we appreciate your time and effort in reviewing our work.
|
||
2. Point-by-point response to Comments and Suggestions for Authors |
||
Comments 1: Page 12 Line 318: The term "hydrogen pressure" should be replaced with "water pressure" for clarity, as the context suggests that water pressure is being discussed.
|
||
Response 1: Thank you for pointing this out. We sincerely apologize for the mistake. We agree with this comment and have replaced the term "hydrogen pressure" with "water pressure". This change has been made in the revised manuscript on Page 12, Line 347. Thank you for your valuable suggestion. |
||
|
||
Comments 2: Page 15 Line 405: The reference to "Figure 13" should be corrected to "Figure 14." Additionally, it would be beneficial that Figure 14 includes the calculated COâ‚‚ conversion limit, which is obtained from the thermodynamic equilibrium limitation in the PBR for better clarity.
|
||
Response 2: Yes, we have made the necessary corrections. The reference to "Figure 13" has been corrected to "Figure 14" on Page 15, Line 405. Additionally, we have included the calculated COâ‚‚ conversion limit in Figure 14 to reflect the thermodynamic equilibrium limitation in the PBR for better clarity. This modification ensures that the discussion aligns more accurately with the equilibrium constraints. Thank you for your valuable suggestion. |
||
|
||
Comments 3: The Peclet number plays a crucial role in the performance of RWGS membrane reactors. Please provide a more detailed explanation of the Peclet number and its relevance to the system under study. |
||
|
||
Response 3: Thank you for your valuable comment. We acknowledge the importance of the Peclet number in determining the performance of RWGS membrane reactors and have accordingly revised the manuscript to provide a more detailed explanation of its significance in our system. Specifically, we have replaced: From this point onwards, the simulation will use actual catalyst and membrane data to replicate the real system, necessitating system validity testing. The analysis confirms that a one-dimensional (1D) model is appropriate for this system, supported by several key parameters. The high Peclet number of 3338.46 demonstrates the dominance of axial flow over diffusion, while the small effective diffusion coefficient (0.0004345 cm²/s) indicates negligible radial diffusion. The system maintains plug flow conditions due to its low flow velocity (1.45 mm/s), and the short reactor length (0.09 m) combined with the high Peclet number ensures axial reaction homogeneity. Additionally, the calculated critical Peclet number (0.099493) is substantially lower than the system's Peclet number, further validating the use of a 1D approach. These conditions collectively confirm that the reactor system can be accurately represented using a simplified one-dimensional model.
To: From this point onwards, the simulation will use actual catalyst and membrane data to replicate the real system, necessitating system validity testing. To ensure an accurate simulation of the real system based on actual catalyst and membrane data, it is crucial to validate whether a one-dimensional (1D) model is suitable for describing the reactor system. The validity of the 1D approach is determined by evaluating several key criteria. First, the dominance of axial flow is confirmed by the significantly high Peclet number (3338.46), which indicates that axial convection far exceeds diffusion. Additionally, radial diffusion is negligible due to the small effective diffusion coefficient (0.0004345 cm²/s), ensuring that mass transfer in the radial direction does not impact the overall performance. The system also satisfies the laminar or plug flow assumption, as evidenced by the low superficial velocity (1.45 mm/s) in the packed bed. Furthermore, axial reaction homogeneity is achieved through the combination of a high Peclet number and a short reactor length (0.09 m), ensuring uniform reaction conditions along the reactor's axial direction. Moreover, radial effects are minimal, as lateral diffusion is small and advection remains the dominant transport mechanism in this system. A critical evaluation based on the Peclet number criterion (Lundin et al.) further supports the use of a 1D model. At 609 K, the critical Peclet number (Pecrit) is calculated as 0.099493, which is much lower than the actual Peclet number (3338.46). Since the system satisfies the condition Pe > Pecrit, the 1D approach is justified, reducing computational complexity while ensuring accurate results. Mathematically, the Peclet number (Pe) is determined in equation 28. The effective diffusion coefficient is determined using equation 29, accounting for catalyst particle size and inner reactor diameter. The critical Peclet number (Pecrit) is derived from an empirical equation, ensuring that the correct transport regime is identified. Data that summarizes the determination of the use of the 1D model can be seen in table 10. In conclusion, all required criteria for applying a 1D reactor model are met. Given that Pe≫Pecrit, the system can be accurately modeled using a 1D approach, providing a computationally efficient yet reliable representation of the reactor behavior. However, if Pe ≤ Pecrit, a 2D model would be necessary to capture the additional transport effects and improve accuracy.
, and added table 10 ensuring a clearer and more comprehensive discussion on how the Peclet number influences convection-diffusion transport, reaction homogeneity, and system validity for the 1D modeling approach. We appreciate your insightful feedback and believe this addition enhances the clarity of our work. |
||
|
||
Comments 4: There are a significant number of equations presented in the manuscript, and they may be confusing to readers. It would be helpful to briefly explain each equation and its purpose within the context of the study to improve readability and understanding. |
||
|
||
Response 4: Thank you for your insightful feedback. We acknowledge that the equations in the "Other Related Equations" section were extensive and could be overwhelming for readers. To improve clarity and organization, we have categorized the equations based on their specific functions in our calculations.
The revised structure groups the equations into the following sections:
Reaction Rate Calculation Permeation Calculation Material Balance Calculation Heat Balance Calculation Heat Transfer Coefficient Equations Thermodynamic Equations Equations for Evaluating the Effect of Water Removal Peclet Number, Critical Peclet Number, and Related Equations Ratio and Membrane Selectivity Equations
These changes ensure that each equation is clearly contextualized, making it easier for readers to understand their relevance to the study. The modifications can be found in the updated equations section of the manuscript.
We appreciate your suggestion, as this restructuring enhances the readability and overall clarity of our work. |
||
|
||
Comments 5: The concept of the Half MPBR is quite interesting and innovative. I suggest further elaborating on its advantages and potential applications in the RWGS process, as this could greatly enhance the impact of the study.
|
||
Response 5: Thank you for your insightful feedback. We appreciate your interest in the Half MPBR concept and its potential impact on the RWGS process. In response to your suggestion, we have incorporated additional data that was previously omitted to further clarify the advantages and potential applications of this design. To support this discussion, we have also included new figures, particularly Figure 24, which provides deeper insights into the performance of the Half MPBR. These revisions can be found in the updated manuscript on page 24. We sincerely appreciate your suggestion, as it has helped strengthen the depth and impact of our study. |
||
|
||
Dear Reviewer 1,
We sincerely appreciate the time and effort you have dedicated to reviewing our manuscript. Your valuable insights and constructive feedback have been instrumental in enhancing the clarity, depth, and overall quality of our study.
With this response, we have carefully addressed all the comments and incorporated the necessary revisions into the manuscript. We have provided detailed explanations and additional data where needed to ensure clarity and completeness.
Your thoughtful comments have allowed us to refine our analysis, strengthen our discussion, and present our findings more effectively. We are truly grateful for your expertise and contribution in improving our work.
Thank you once again for your careful evaluation and for helping us improve this manuscript.
Best regards, Author |

Reviewer 2 Report
Comments and Suggestions for Authors
This paper has potential to be a good paper. It is however not very well presented. It resembles more a laboratory report or term paper at a university than a solid scientific paper. It needs major improvements to reach the minimum quality for publication. It is not necessary to present every little experiment that was performed, which has the consequence of increasing the number of graphs and tables. I do not recommend accepting this paper in its current format.
The abstract is a good summary of the content of the paper. However, it is a bit general and could be enhanced by adding a few actual numbers of the comparison in performance.
(line 36) I am not sure of the way used by the authors to reference papers is the right one. The authors used (1–5) and, according to the instructions for authors, it would be preferable to use [1-5]
Is it a good idea to use the Reverse Water Gas Shift (RWGS) reaction to remove carbon dioxide instead of using hydrogen as a fuel, especially when it usually needs to be produced at a relatively high cost and, very often, from natural gas? Maybe a few sentences in the introduction to at least present the context could improve the paper.
I find that the set of equations in the model is not well presented and just dumped in the paper without much information. I think it would be better to properly write this section and put it in an annex at the end of the paper. It is better presented in Zhuang et al. (Ref. 29).
(line 167) It is written: “Subsequently, the parameters used in the initial journal were modified to achieve more optimal simulation results.” I am not sure about the meaning of this statement, as the parameters were validated in reference 29. Were the parameters modified based on actual experiments? One can always change model parameters, but it needs to be supported with experimental results. This statement needs to be clarified.
(Table 1) Maybe “heat of adsorption” would be a better term to use. In addition, adsorption is typically exothermic, why is one positive and the other one negative? Are the catalyst properties determined by the authors or taken from a paper? In the latter, a reference should be provided.
(Table 3) What is the variable C [K]? It does not appear in the nomenclature.
(line 176) Subscript 2
(line 188) Change “… we will examine the design and comparison of different reactor …” to “… we will examine and compare the design of different reactor …”
I am not sure if Figure 2 is necessary in this paper. The text is sufficiently descriptive and fairly obvious.
(Table 6) Change to “Experimental conditions”. Change pa to Pa for the three appearances of this unit.
(Figure 5) The authors use ml and mL at different points in the paper. I suggest using only one.
(line 305) In Table 8, change (1) pa for Pa and (2) the temperature for Temperature. In addition, the “±” is confusing and it would probably be better to use “~” to mean approximately. In addition, the term “parameter” is misleading as the authors are really referring to “process variables”.
(Figure 6) The presence of oscillation in the temperature graph (6c) is surprising for simulated results. In addition, the caption should be improved and more descriptive.
(line 345) The number of points used to draw each plot in Figure 8 is too small, which makes the plots lack smoothness.
I am not sure of the validity of the results of Figure 8. Even if all the water is removed, the conversion may not reach 100% as CO is also part of the equilibrium reaction. This should be briefly discussed.
(line 433) The number of points used to draw each plot in Figure 15 is too small, which makes the plots lack smoothness.
The reference section needs some attention, such as the subscripts of the numerous molecular formulas.
Comments on the Quality of English LanguageGood
Author Response
For research article
Response to Reviewer 2 Comments
|
||
1. Summary |
|
|
Dear Reviewer,
Thank you very much for taking the time to review our manuscript. We sincerely appreciate your valuable feedback and thoughtful suggestions. Please find our detailed responses below, along with the corresponding revisions and corrections, which have been highlighted in track changes in the re-submitted files. We truly appreciate your effort in evaluating our work and helping us improve the quality of this study.
Best regards, Authors
|
||
2. Point-by-point response to Comments and Suggestions for Authors |
||
Comments 1: The abstract is a good summary of the content of the paper. However, it is a bit general and could be enhanced by adding a few actual numbers of the comparison in performance.
|
||
Response 1: Thank you for your valuable feedback. We have revised the abstract to include specific numerical comparisons in performance, as suggested. The updated abstract can be found in the revised manuscript. |
||
|
||
Comments 2: (line 36) I am not sure of the way used by the authors to reference papers is the right one. The authors used (1–5) and, according to the instructions for authors, it would be preferable to use [1-5] |
||
|
||
Response 2: Thank you for your comment. The citation format has been corrected to [1-5] in accordance with the journal's instructions. Additionally, we have ensured that all formatting follows the guidelines of the Membranes journal. The revised version can be found in the updated manuscript. We appreciate your careful review. |
||
|
||
Comments 3: Is it a good idea to use the Reverse Water Gas Shift (RWGS) reaction to remove carbon dioxide instead of using hydrogen as a fuel, especially when it usually needs to be produced at a relatively high cost and, very often, from natural gas? Maybe a few sentences in the introduction to at least present the context could improve the paper. |
||
|
||
Response 3: Thank you for your insightful comment. We acknowledge that the Reverse Water Gas Shift (RWGS) reaction requires hydrogen, which is often produced from natural gas and can be costly. However, RWGS is a promising approach for COâ‚‚ utilization, as it converts COâ‚‚ into CO, which can serve as a key feedstock for synthesis gas (syngas) production, leading to valuable fuels and chemicals.
Additionally, with the growing development of green hydrogen production from renewable energy sources, RWGS has the potential to become a more sustainable route for carbon utilization. To address this, we have added a brief discussion in the introduction to provide better context on the feasibility and potential applications of RWGS in COâ‚‚ utilization strategies. we also added new references (now 29-32) to strengthen this statement.
We appreciate your valuable feedback, as it helps enhance the clarity and relevance of our study. |
||
|
||
Comments 4: I found that the set of equations in the model was not well presented and just dumped in the paper without much information. I think it would be better to properly write this section and put it in an annex at the end of the paper. It is better presented in Zhuang et al. (Ref. 29). |
||
|
||
Response 4: Thank you for your feedback. We have revised the presentation of the equations to improve clarity and organization. The changes have been made accordingly, and the updated section can be reviewed in the manuscript. |
||
|
||
Comments 5: (line 167) It is written: “Subsequently, the parameters used in the initial journal were modified to achieve more optimal simulation results.” I am not sure about the meaning of this statement, as the parameters were validated in reference 29. Were the parameters modified based on actual experiments? One can always change model parameters, but it needs to be supported with experimental results. This statement needs to be clarified. |
||
|
||
Response 5: Thank you for your comments. The modifications made in the simulation were based on a validated experimental dataset, ensuring that the model accurately represents real experimental conditions. The initial parameters were taken from previous experimental studies and the model was validated before being used for the modification.
Once the model was confirmed to reliably reflect the experimental system, certain parameters such as temperature, pressure, and flow rates were adjusted within the simulation to explore optimal operating conditions. These modifications were done solely within the simulation environment while maintaining consistency with the validated experimental data.
We have clarified this statement in the manuscript to better reflect the methodology used. Thank you again for your valuable feedback.
initial sentence: Subsequently, the parameters used in the initial journal were modified to achieve more optimal simulation results. This optimization will serve as a foundation for further simulations, aiming to improve the accuracy and effectiveness of the catalytic system.
change: The simulation was developed using validated experimental data to ensure accuracy in representing real conditions. Once the model was validated, parameters such as temperature, pressure, and flow rates were adjusted to explore optimal conditions. This optimization provides a foundation for further simulations to enhance catalytic system performance. |
||
|
||
Comments 6: (Table 1) Maybe “heat of adsorption” would be a better term to use. In addition, adsorption is typically exothermic, why is one positive and the other one negative? Are the catalyst properties determined by the authors or taken from a paper? In the latter, a reference should be provided. |
||
|
||
Response 6: Thank you for your insightful comment. We have used this terminology to remain consistent with the referenced journal. To clarify this, we have added the appropriate citation in the revised manuscript.
We appreciate your valuable feedback and careful review. |
||
|
||
Comments 7: (Table 3) What is the variable C [K]? It does not appear in the nomenclature. |
||
|
||
Response 7: Thank you for your comment. The variable C [K] refers to the Sutherland constant. We have clarified this in the manuscript to ensure consistency and completeness in the nomenclature section. We appreciate your careful review and valuable feedback. |
||
|
||
Comments 8: (line 176) Subscript 2 |
||
|
||
Response 8: Thank you for your careful review. The correction has been made accordingly. We appreciate your attention to detail. |
||
|
||
Comments 9: (line 188) Change “… we will examine the design and comparison of different reactor …” to “… we will examine and compare the design of different reactor …” |
||
|
||
Response 9: Thank you for your suggestion. The revision has been made accordingly in the manuscript. We appreciate your careful review. |
||
|
||
Comments 10: I am not sure if Figure 2 is necessary in this paper. The text is sufficiently descriptive and fairly obvious. |
||
|
||
Response 10: Thank you for your feedback. While the text provides a detailed explanation, we believe that Figure 2 remains valuable as a visual aid for readers who may not be as familiar with this concept. Retaining the figure ensures accessibility for a broader audience, particularly those who are still learning and may find a visual representation helpful. We hope this approach makes sense. |
||
|
||
Comments 11: (Table 6) Change to “Experimental conditions”. Change pa to Pa for the three appearances of this unit. |
||
|
||
Response 11: Thank you for your suggestion. The requested changes have been made accordingly. We appreciate your careful review. |
||
|
||
Comments 12: (Figure 5) The authors use ml and mL at different points in the paper. I suggest using only one. |
||
|
||
Response 12: Thank you for your suggestion. We have standardized the unit notation to mL throughout the manuscript for consistency. We also check the other unit notation to prevent the same mistake. We appreciate your careful review. |
||
|
||
Comments 13: (line 305) In Table 8, change (1) pa for Pa and (2) the temperature for Temperature. In addition, the “±” is confusing and it would probably be better to use “~” to mean approximately. In addition, the term “parameter” is misleading as the authors are really referring to “process variables”. |
||
|
||
Response 13: Thank you for your valuable feedback. We have made the necessary changes as suggested to improve clarity and consistency in Table 8. We appreciate your careful review and insightful suggestions. |
||
|
||
Comments 14: (Figure 6) The presence of oscillation in the temperature graph (6c) is surprising for simulated results. In addition, the caption should be improved and more descriptive. |
||
|
||
Response 14: Thank you for your insightful comment. The slight oscillation observed in the temperature profile (Figure 6c) is likely due to localized thermal effects at the reactor inlet. The reaction is endothermic reaction, and the reaction rate is highest near the inlet. That’s why the fluctuation is appear only near the inlet. At this point, the rapid COâ‚‚ hydrogenation could cause minor temperature fluctuations before reaching a steady-state profile. However, this effect is compensated by continuous heating from the reactor walls, leading to overall temperature stabilization further along the reactor.
Additionally, most of the reactor exhibits a stable temperature profile, with oscillations occurring only in a small region near the inlet. In experimental setups, temperature sensors are typically placed in the middle of the packed bed, which may explain why this fluctuation is not commonly observed in measured data. The observed oscillations remain within a narrow range, indicating that they do not significantly affect system performance.
At this stage, we don`t know the reason yet, but the possibility of a numerical problem is quite there. with the program we are using now (FlexPDE) we get the following results, and we have not tried with other systems. in the future we may try to compare from various systems to get an answer whether it is true that the results are true oscillations or numerical problems.
To clarify this, we have also improved the Figure 6 caption in the manuscript. Thank you again for your valuable feedback. an explanation of this will also be added to the manuscript.
Revised Caption for Figure 6: Figure 6. (a) COâ‚‚ conversion profile in different reactor configurations (PBR, MPBR, and MPBR with permeation inlet). (b) Water pressure distribution along the reactor for the reaction and permeation sides. (c) Temperature profile along the reactor, where minor oscillations at the inlet region are observed due to localized reaction effects before reaching thermal stability. |
||
|
||
Comments 15: (line 345) The number of points used to draw each plot in Figure 8 is too small, which makes the plots lack smoothness. |
||
|
||
Response 15: Thank you for your feedback. We have increased the number of data points in Figure 8 to improve the smoothness of the plots. The updated figure can be found in the revised manuscript. We appreciate your careful review. |
||
|
||
Comments 16: I am not sure of the validity of the results of Figure 8. Even if all the water is removed, the conversion may not reach 100% as CO is also part of the equilibrium reaction. This should be briefly discussed. |
||
|
||
Response 16: Thank you for your insightful comment. You are correct that the conversion would not reach exactly 100%, as CO remained part of the equilibrium reaction. In our simulation, water removal occurs through membrane permeation, which is driven by partial pressure differences, meaning that not all water can be completely removed. This is reflected in Figure 12, where the conversion does not reach 1 but instead approaches 0.99, demonstrating the practical limitations of the process.
In Figure 8, the assumption of complete water removal is used as a theoretical reference to illustrate the maximum possible shift in equilibrium. While this suggests a conversion of 1 in an ideal case, it serves as an upper bound rather than a practically achievable result.
To clarify this point, we have added a brief discussion in the revised manuscript. We appreciate your thoughtful feedback and your careful review of our work.
additionally inputted to the manuscript near figure 8:
‘Figure 8 illustrates the theoretical upper limit of conversion assuming complete removal of water from the system. However, water removal occurs through membrane permeation, which is driven by partial pressure differences. As a result, complete water removal cannot be practically achieved, and the conversion does not reach exactly 1. This is evident in Figure 12, where the simulated conversion is closer to 0.99 than 1, demonstrating the inherent limitations of the process. Therefore, although Figure 8 serves as a reference to illustrate the maximum possible equilibrium shift, the actual performance of the system is limited by the ability of the membrane to selectively remove water under realistic operating conditions.'
addition inputted to the manuscript near figure 12: ‘As shown in Figure 12, the conversion approaches 0.99 instead of 1, confirming that the system is still governed by practical constraints in water removal through membrane permeation.’ |
||
|
||
Comments 17: (line 433) The number of points used to draw each plot in Figure 15 is too small, which makes the plots lack smoothness. |
||
|
||
Response 17: Thank you for your feedback. We have increased the number of data points in Figure 15 to improve the smoothness of the plots. The updated figure can be found in the revised manuscript. We appreciate your careful review. |
||
|
||
Comments 18: The reference section needs some attention, such as the subscripts of the numerous molecular formulas. |
||
|
||
Response 18: Thank you for your comment. We have carefully revised the reference section and corrected the subscripts in the molecular formulas as needed. We appreciate your attention to detail and your valuable feedback. |
||
|
||
3. Additional clarifications |
||
Dear Reviewer,
Thank you for your feedback. We acknowledge your concerns regarding the structure and presentation of the paper. The current format was developed based on editorial guidance, as we were advised to expand the content to ensure sufficient depth and completeness.
To clarify, we initially received feedback from the editorial board indicating that the main content was too short, and we were encouraged to extend the manuscript to provide a more comprehensive discussion. Below is the email:
“……The editorial board member has checked this manuscript. We found that the main content was too short. We encourage authors to extend more content in the main text. Regarding the word number, it is not a hard limitation but a suggested one. The aim is to make sure that the paper has enough and rich content, and we hope to publish good papers. ….”
In response to this, we expanded the manuscript by including additional data, figures, and discussions to meet the editorial requirements while maintaining scientific rigor. However, we understand the importance of concise and well-structured presentation, and we will carefully refine the manuscript to improve clarity and readability.
We sincerely appreciate your constructive feedback and will make the necessary improvements to enhance the overall quality of the paper. Thank you again for your time and thoughtful review.
Best regards, Authors |

Round 2
Reviewer 2 Report
Comments and Suggestions for Authors
I have reviewed the new version of the paper as well as the responses to my first review. The authors have addressed adequately most of the comments. The paper is a much better version and I recommend the publication of this paper even though I do not completely agree entirely with some of the answers. The authors has enhanced sufficiently the paper for its publication.